# High-strength and machinable load-bearing integrated electrochemical capacitors based on polymeric solid electrolyte

Jinmeng Zhang[1,3], Jianlong Yan[1,3], Yanan Zhao[1], Qiang Zhou[1], Yinxing Ma[2], Yaxian Zi[1], Anan Zhou[1], Shumin Lin[1], Longhui Liao[1], Xiaolan Hu [1]✉ & Hua Bai [1]✉

Load bearing/energy storage integrated devices (LEIDs) allow using structural parts to store energy, and thus become a promising solution to boost the overall energy density of mobile energy storage systems, such as electric cars and drones. Herein, with a new high-strength solid electrolyte, we prepare a practical high-performance load-bearing/energy storage integrated electrochemical capacitors with excellent mechanical strength (flexural modulus: 18.1 GPa, flexural strength: 160.0 MPa) and high energy storage ability (specific capacitance: 32.4 mF cm$^{-2}$, energy density: 0.13 Wh m$^{-2}$, maximum power density: 1.3 W m$^{-2}$). We design and compare two basic types of multilayered structures for LEID, which significantly enhance the practical bearing ability and working flexibility of the device. Besides, we also demonstrate the excellent processability of the LEID, by forming them into curved shapes, and secondarily machining and assembling them into complex structures without affecting their energy storage ability.

Load bearing/energy storage integrated devices (LEIDs) refer to multifunctional structural devices with both mechanical bearing capacity and electrochemical energy storage capacity[1–3]. In conventional power supply mode, the energy storage and load-bearing components are independent. The power storage component can store energy but cannot withstand large external forces, while the load-bearing components, such as the shell, can only play the role of protection and support and cannot provide energy storage[4–6]. However, the total energy density of a system is defined as the ratio of the total energy stored in it to its total mass[4,6–11]. From the perspective of energy storage, load-bearing components in conventional power supply systems can be defined as dead mass[7], which reduces the total energy density of the system[4,6,12,13]. For example, in an electric car, the metal shell and chassis are dead mass because they make up 40% of the car's weight, but they have zero energy density[7]. If the energy-storage component has sufficient strength and can serve as mechanical support, it can replace the structural component. For the whole system, the total energy density is increased because the usage of dead mass can be

reduced[14,15]. With LEIDs as shells, satellites, electric vehicles, drones, laptops, and mobile phones can have more electric power and longer battery life. Besides, LEIDs can also serve as support structures and energy storage units for intermittent new energy sources, such as wind power and photovoltaics. Consequently, LEIDs significantly increase the energy density of mobile energy storage systems and simplifies the system[16].

A LEID requires a unique design on the device structure and material level. The electrodes and the solid electrolyte of the device should have both high mechanical strength and good interface bonding[17–19]. Currently, LEIDs are mainly designed based on high-strength resin-based carbon fiber composite materials[20,21], where the carbon fiber (CF) with high strength and good electrical conductivity is the reinforcement and the electrodes, and the high-strength solid electrolyte serves the matrix to bond CFs together[2,22]. However, although the concept of LEID has been proposed for decades[4,5,23,24], there is few practical LEID reported in literature[20,25]. One reason for this situation is the lack of solid electrolytes with high mechanical strength,

[1]College of Materials, Xiamen University, Xiamen 361005, PR China. [2]College of Chemistry and Chemical Engineering, iChEM, Xiamen University, Xiamen 361005, PR China. [3]These authors contributed equally: Jinmeng Zhang, Jianlong Yan. ✉e-mail: xlhu@xmu.edu.cn; baihua@xmu.edu.cn

high ionic conductivity, and good processability. The soft solid electrolyte developed for lithium-ion batteries usually cannot meet the requirement of LEID[23,26,27]. Besides, the structure of the device has not been purposefully designed and studied. A limited number of high-strength polymeric solid electrolytes were only tested in a simplified model device with a thin three-layer CF/separator/CF structure, and the mechanical properties of the devices were seldom mentioned[4,20]. These thin and simple three-layered devices are far from real LEIDs, which should be mechanically strong and easy to process. To develop a LEID aiming at future practical application, we need to prepare multilayered LEID compatible with the forming/processing methods of conventional resin-based composites, and evaluate their mechanical properties following the standards of composite materials. Furthermore, it is also necessary to understand the coupling relationship between mechanics/energy storage performance, such as the effect of mechanical damage on the energy storage performance of the device[28].

In this paper, we demonstrate a practical high-performance LEID with a newly developed high-strength polymeric solid electrolyte. For the first time, the multilayered structure was designed for the LEID, and two basic multilayered structures were investigated and compared. The multilayered structure allows us to construct thicker and stronger LEIDs, and it also provides flexible ways of using the LEID as the power supply. The LEIDs exhibited large energy capacity and high mechanical strength/toughness superior to common engineering plastics, indicating their practicability as both the power source and structural material. We also demonstrate that the LEIDs can be made into complex shapes composed of curved surfaces and are capable of secondary machining. After the device was cut, drilled, and assembled, its energy storage performance was not decreased. Therefore, these practical LEIDs can be easily used as parts and integrated into a complex system.

## Results

### Preparation and characterization of high-strength solid electrolyte

High-strength solid electrolyte is essential for LEID, but the conventional solid electrolyte based on thermoplastic polymers, such as poly(ethylene oxide) (PEO) and poly(vinylidene difluoride) (PVDF), do not have sufficient high mechanical strength[29,30]. Therefore, solid electrolytes based on high-strength thermosetting polymers become suitable candidates. However, the thermosetting polymer has a rigid molecular chain network and small free volume after cross-linking; thus, the transport of ions through the polymer matrix is difficult. Consequently, the solid electrolyte should be designed as a bi-continuous-phase type structure, where the liquid electrolyte forms a separated phase inside the porous polymer matrix, so that the ion transport is not restricted by polymer chains. Here in this work, we developed a new bi-continuous-phase type solid electrolyte[31]. Epoxy resin 5284 was chosen as the matrix phase, while the solution of bis(trifluoromethane)sulfonamide lithium (LiTFSI) in 1-ethyl-3-methylimidazolium bis(trifluoromethanesulfonyl)imide (EMIm-TFSI) ionic liquid was used as the conductive phase (Fig. 1a). Epoxy resin 5284 is a single-component epoxy resin with cyanate resin as the curing agent[32–34]. The bisphenol A type cyanate resin has a rigid molecular structure, thereby bringing a strong cross-linking network into the cured resin, making the resin show outstanding mechanical strength and thermal stability. Therefore, epoxy resin 5284 is expected to provide high mechanical properties for solid electrolytes. EMIm-TFSI was used as the electrolyte because it was reported to undergo a spontaneous phase-separation during the curing of epoxy resin[14]. The solution of LiTFSI in EMIm-TFSI (LE) has high ionic conductivity (2.1 mS cm$^{-1}$), comparable to the commercial liquid electrolyte in the lithium-ion battery, so it can ensure a good conductivity of the solid electrolyte (Supplementary Table 2). As shown in Fig. 1a, the high-

strength solid electrolyte is prepared via crosslink-induced phase-separation[14,26]. A liquid electrolyte (LE) containing LiTFSI dissolved in EMIm-TFSI was mixed with epoxy resin 5284 liquid precursor to form a uniform and transparent mixed liquid, which was then cured at a certain temperature. Phase separation between LE and epoxy resin occurred during curing, producing a bicontinuous-phase type solid electrolyte. The mass ratios of the epoxy resin and LE were 100:0, 70:30, 60:40, 55:45, 50:50, 45:55, and 40:60 (Supplementary Table 1), and the cured products obtained are denoted as $EP_{100}$, $EP_{70}$, $EP_{60}$, $EP_{55}$, $EP_{50}$, $EP_{45}$, and $EP_{40}$, respectively.

The curing process was first investigated by differential scanning calorimetry (DSC, Supplementary Fig. 1) and gel point tests (Supplementary Fig. 3) to determine the curing temperature, and the successful curing was finally verified by Fourier transform infrared (FT-IR) spectra (Supplementary Fig. 4). However, the measurement of the curing enthalpy suggests that the crosslinking density of epoxy resin decreased after LE was added (Supplementary Fig. 1), because LE reduced the concentration of epoxy groups. Figure 1b shows the photos of the pure epoxy resin and solid electrolytes[26,35]. All the samples are hard solids. With the addition of LE, the sample gradually changed from transparent to opaque white, indicating the formation of microstructure inside the sample. Figure 1c, d show the microscopic morphologies of $EP_{70}$ and $EP_{50}$ (The morphologies of other samples are shown in Supplementary Fig. 5), and LE had been removed from the samples by ethanol extraction before the scanning electron microscopy (SEM) observation. The cross-section of the $EP_{70}$ sample is dense, indicating that LE formed a homogeneous phase with epoxy resin. The dynamic thermomechanical analysis (DMA) showed that the glass transition temperature ($T_g$) of the samples was reduced from 179 °C of pure epoxy resin ($EP_{100}$) to 129 °C of $EP_{70}$ (Supplementary Fig. 6), which reveals that LE plasticized the epoxy resin[13,36–38]. When the LE content reaches 45% ($EP_{55}$), some isolated pores can be observed in the SEM image (Supplementary Fig. 5), showing that phase separation between LE and epoxy occurred. This phase separation is further confirmed by the DSC measurement (Supplementary Fig. 2), in which two glass transition temperatures were observed. The cross-linking of epoxy resin reduces the solubility of LE in epoxy resin and causes the formation of the LE phase. Further increasing the content of LE to higher than 50%, the spinodal decomposition occurred, producing a bi-continuous structure, as observed in SEM images (Fig. 1d and Supplementary Fig. 5). The domain size of the bi-continuous structure increases significantly with the LE content, from 120 ± 20 nm ($EP_{50}$) to 400 ± 40 nm ($EP_{45}$) and 560 ± 55 nm ($EP_{40}$). It is noteworthy that the $T_g$ of the above phase-separated samples continued to decrease with the increased LE content, indicating that the epoxy resin phase was always plasticized by LE in the solid electrolytes (Supplementary Fig. 6). Therefore, the solid electrolytes have a bi-continuous structure composed of a LE phase and a LE-plasticized epoxy resin phase[37,39,40].

The mechanical properties and ionic conductivity of solid electrolytes were investigated. The $EP_{100}$ sample breaks before yielding with an elongation at break of only 4.6%, showing typical brittle fracture behavior. After LE was added, the toughness of the samples was improved significantly. The tensile curves (Fig. 1e) show that all the samples containing LE yielded before the break, and the elongation at break also increased. The impact strength of the sample increased from 13.0 kJ m$^{-2}$ of $EP_{100}$ to 19.0 kJ m$^{-2}$ of $EP_{55}$ electrolyte, confirming that LE increases the toughness of the solid electrolyte (Supplementary Fig. 7). As discussed above, LE effectively plasticized the epoxy resin phase, so the toughness of the solid electrolyte increased with the LE content. However, the breaking strength and modulus of the sample also decreased with the increase of LE content. The tensile modulus and strength of the $EP_{100}$ sample were 2.5 GPa and 78.2 MPa, but the tensile modulus and strength of $EP_{50}$ dropped to 1.0 GPa and 11.6 MPa (Fig. 1e, g). The results of bending tests showed a similar trend (Supplementary Fig. 7). Actually, the mechanical strength of the solid

electrolyte is mainly provided by the epoxy resin phase, so the decrease in the volume fraction of the epoxy resin phase leads to smaller strength and modulus of the solid electrolyte.

In contrast to the mechanical strength, the ionic conductivity of the solid electrolyte increased with the LE (Fig. 1f, g and Supplementary Fig. 8). When the LE content is less than 45%, no continuous LE phase is formed, and the ions between the molecular chains have low mobility, so the conductivities of the solid electrolytes are very low. The conductivity of $EP_{60}$ is only $0.9 \times 10^{-2}$ mS cm$^{-1}$, while the conductivity of $EP_{70}$ and $EP_{100}$ is too low to be measured. As the content of LE increases, the ionic conductivity of $EP_{50}$ becomes $6.7 \times 10^{-1}$ mS cm$^{-1}$, and for $EP_{40}$, the conductivity can reach $9.5 \times 10^{-1}$ mS cm$^{-1}$. This con-

ductivity is much higher than those of solid electrolytes with similar mechanical strength and can be ascribed to the well-developed LE phase. For LEIDs, both the mechanical strength and the ionic conductivity are essential. To achieve a balance between mechanical properties and electrical conductivity, we used $EP_{50}$ as the solid electrolyte in the subsequent research. Its tensile modulus, tensile strength, and ionic conductivity are 1.0 GPa, 11.6 MPa, and $6.7 \times 10^{-1}$ mS cm$^{-1}$, respectively. Fig 1h compares the performance of $EP_{50}$ in this work and the literature data, and the data point of $EP_{50}$ is located in the upper right corner of the figure, indicating that its comprehensive performance is better than the literature value[14,37,39–42] (Supplementary Table 3).

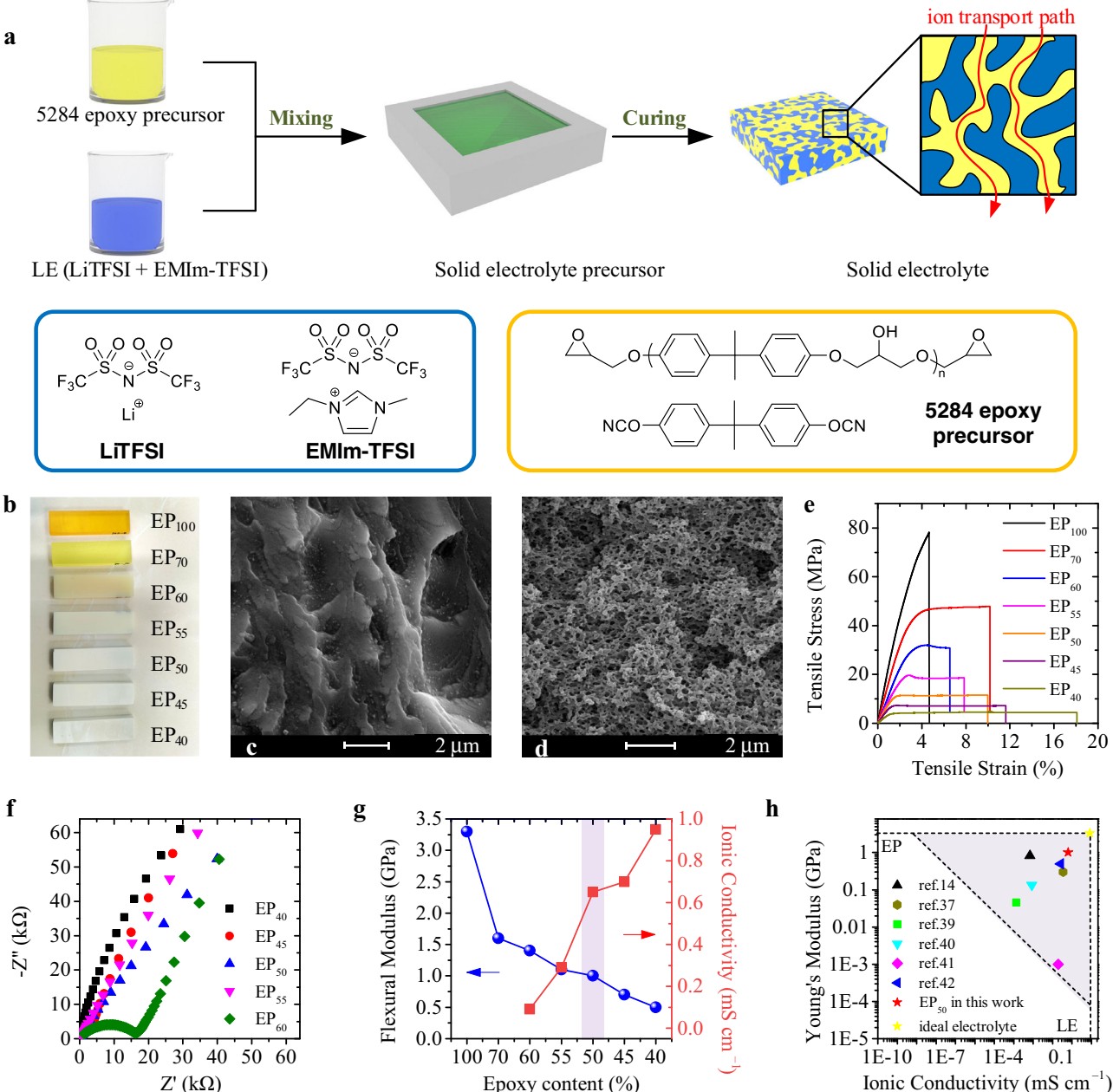

**Fig. 1 | Structure and properties of the epoxy resin-based solid electrolyte.**
**a** Schematic diagram of the preparation of solid electrolyte with epoxy resin and ionic liquid, and the chemical structures of the epoxy resin and the ionic liquid. **b** Photos of epoxy resin with different ionic liquid content. **c, d** SEM images of $EP_{70}$ (c) and $EP_{50}$ (d). **e** Tensile stress-strain curves of epoxy resin and different solid electrolytes. **f** Impedance spectra of different solid electrolytes. **g** The Flexural modulus (left axis) and ionic conductivity (right axis) of different samples at 25 °C. **h** Comparison of the solid electrolytes in literature with $EP_{50}$ in this work.

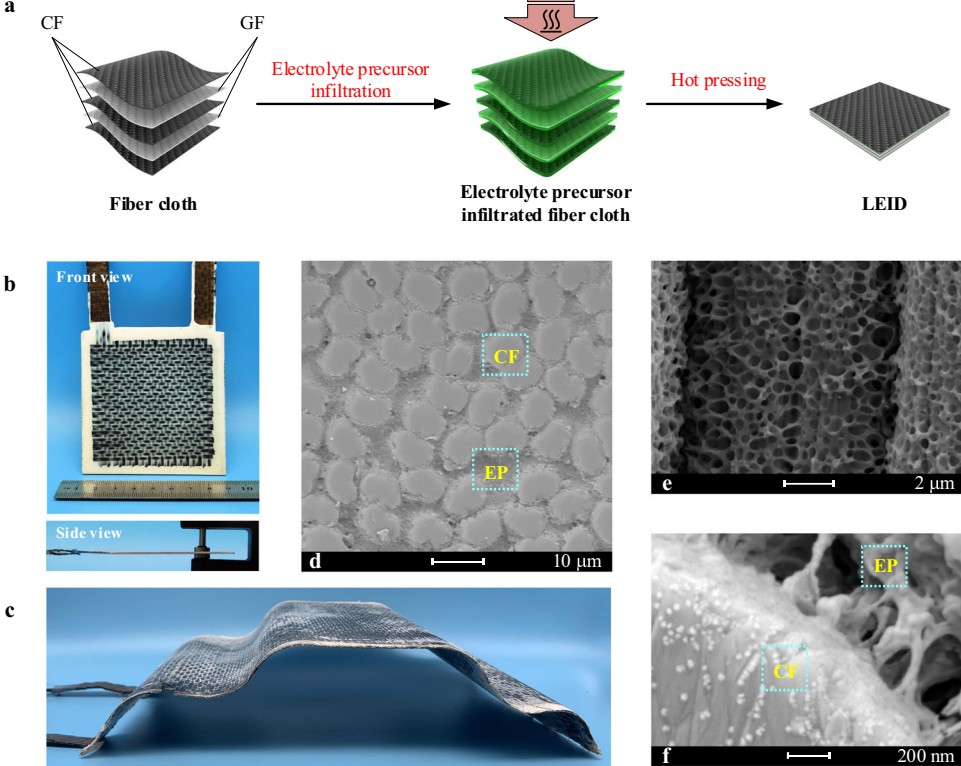

**Fig. 2 | Preparation and structure of LEID. a** Schematic diagram of the preparation of LEID. **b** Front view and side view of a supercapacitor with three-layered structure plate. **c** Image of a car model shell made of LEID-3. **d** SEM image of LEID-3 showing the cross-section of CFs and EP$_{50}$. **e** SEM image of LEID-3 showing the bicontinuous phase of EP$_{50}$ on the surface of CFs. **f** SEM images of the interface between CF and EP$_{50}$.

## Fabrication and characterization of the three-layer load-bearing/energy-storage integrated device

LEIDs were then fabricated with EP$_{50}$ as the solid electrolyte using a traditional prepreg-molding process (Fig. 2a)[43–46], which can be easily scaled up in the industry. The structure of the LEID is the same as that of resin-based composite. As shown in Fig. 2a, CF fabrics were used as the electrodes, and glass fiber (GF) fabrics as the separators to avoid short circuits, and both CF fabrics and GF fabrics also served as the mechanical reinforcements. A LEID with a conventional three-layered structure (LEID-3) was first constructed with two pieces of CF fabric as electrodes and a piece of GF fabric as the separator. Fig 2b shows the photo of a LEID-3 with a platelike shape. The device is a rigid plate with a thickness of 1.2 mm, and the volume fraction of the fiber is 38%-43% (Supplementary Table 4). It is worth noting that our LEID can be fabricated into various shapes with proper molds (Supplementary Fig. 9). A LEID-3 car shell model is shown in Fig. 2c (For its electrochemical properties, see Supplementary Fig. 10). The size of the model is 24.2 cm in length, 14.8 cm in width, and 1.2 mm in thickness, with an area of 358 cm$^2$. The curved and planar regions in the model are uniform, demonstrating the excellent molding ability of our LEID, which can meet the demand of practical applications (Supplementary Movie 1). SEM images depict that the EP$_{50}$ solid electrolyte infiltrated into CF and GF fabrics and filled all the space inside and between these fabrics (Fig. 2d), showing a high affinity between CF/GF and the liquid precursor. The EP$_{50}$ in the LEID has a uniform bi-continuous phase structure similar to that of the pure EP$_{50}$, but the pores were larger in LEID-3 than in pure EP$_{50}$, possibly because the CF with high thermal conductivity changed the curing kinetics. The magnified SEM image of the fiber surface shows that the epoxy resin phase in EP$_{50}$ forms good contact with the CF. Meanwhile, some holes penetrate directly to the surface of the CF, indicating that the LE phase is also in contact with the CF (Fig. 2e, f). Therefore, the interface between EP$_{50}$ and CF also has a

two-dimensional bi-continuous pattern. The contact between the epoxy resin phase and the CF endows the LEID with high interface mechanical strength, while the contact between the LE phase and the carbon fiber can realize the energy storage function.

Although the mechanical properties of many solid electrolytes have been reported in the literature, the mechanical performance of LEID has not been studied systematically. Therefore, we first investigated LEID-3's mechanical properties. As shown in Fig. 3a, LEID-3 has significant rigidity, and a weight of 10 kg caused no apparent bending on suspended LEID-3. Fig 3b compares the mechanical properties of the LEID-3 and the pure epoxy-based composite material of the same structure ((CF/GF/CF)$_{EP100}$, CM-3). CM-3 showed a linear stress-strain curve, presenting brittle fracture characteristics with a flexural modulus of 23.9 GPa and flexural strength of 443.0 MPa. LEID-3 yielded when it was bent, showing an improved toughness. The flexural modulus and flexural strength of the LEID-3 are 18.1 GPa and 160.0 MPa, respectively. The modulus of LEID-3 is decreased only by 24.2% compared with that of CM-3, because the flexural modulus of composite materials is mainly determined by reinforcing fibers[17,47,48]. Although the strength of LEID-3 is lower than that of CM-3, it is still significantly larger than those of commonly used engineering plastics (Fig. 3d), such as Nylon 6 (110.0 MPa), polysulfone (107.0 MPa), and acrylonitrile butadiene styrene copolymer (ABS, 76 MPa)[44,49–51]. Therefore, our LEID with high mechanical strength and can be used as the structural material.

Impact performance is essential for the practical applications of composite materials. Here we measured the interlaminar shear strength of LEID-3 and CM-3 (Fig. 3c and Supplementary Fig. 23) to compare their impact resistance. The load-distance curve of CM-3 presents a linear shape, revealing an interlayer shear failure mode. Thus, CM-3, like many other fiber-reinforced epoxy resin matrix composites, suffers from brittleness and low impact resistance.

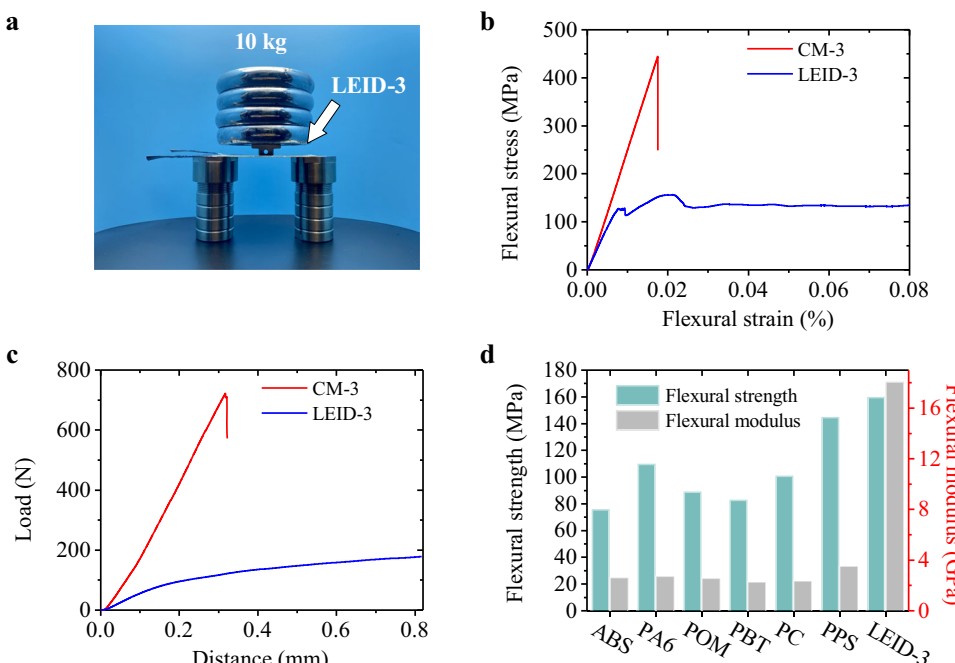

**Fig. 3 | Mechanical properties of LEID-3. a** Photo of a LEID-3 supporting a 10 kg weight. **b** Bending stress-strain curves of CM-3 and LEID-3. **c** Load-distance curves in interlaminar sheet test of CM-3 and LEID-3. **d** Comparison of the mechanical properties of LEID-3 and some engineering plastics.

However, LEID-3 yielded and did not break in the experimental distance range, and it also maintained a certain load-bearing capacity after yielding. These data show that LEID-3 has higher impact toughness than CM-3. The interlaminar shear strengths of CM-3 and LEID-3 are 58.0 MPa and 13.5 MPa, respectively. For LEID-3, due to the high porosity and low strength of the $EP_{50}$ matrix, local squeezing and yielding are more likely to occur, so its interlaminar shear strength is lower than CM-3. However, because the toughness of LEID-3 is significantly higher than that of CM-3, LEID-3 has higher impact resistance.

The electrochemical properties of the LEID-3 were then investigated. The cyclic voltammetry (CV) curves at different scanning rates in the voltage range of 0–2 V all have a deformed rectangular shape without pronounced redox peaks, showing a typical capacitive behavior (Fig. 4a). Figure 4b is the cyclic galvanostatic charge-discharge (GCD) curves of LEID-3. There is no obvious platform in the GCD curves, consistent with the CV curves. According to the GCD curves, it can be calculated that at a current density of 0.2 mA cm$^{-2}$, the areal specific capacitance of LEID-3 is 32.4 mF cm$^{-2}$ (Fig. 4c, 675.9 mF g$^{-1}$, 1297.8 mF cm$^{-3}$, see Supplementary Fig. 11), surpassing those of other solid electrochemical capacitors with CF as the electrode (Supplementary Table 5, Supplementary Fig. 11). Because no electrochemically active materials were used in the LEID, the above capacitance is originated from the electric double-layer capacitance. The LE phase is in contact with the carbon fiber in the LEID, so energy can be stored in the electric double-layer between LE and CF. With the increase of current density, the specific capacitance of the device drops slightly because of the thick solid electrolyte, which limits the ion transport speed. The energy density and power density are calculated to be 0.13 Wh m$^{-2}$ (3.4 × 10$^{-2}$ Wh kg$^{-1}$, 144.9 Wh cm$^{-3}$) and 1.3 W m$^{-2}$ (0.34 W kg$^{-1}$, 1453.3 W cm$^{-3}$). The Ragone plot (Fig. 4d, e, Supplementary Fig. 12) shows that the energy densities of LEID-3 are higher than other solid devices with similar structures and electrode materials in the literature[18,20,21,25,46,52–57]. These data are even higher than those of devices using electroactive material-modified CF as electrodes[6]. The high energy density is ascribed to the large capacitance and high working voltage of LEID-3. After charging the device at a current density of

0.2 mA cm$^{-2}$ for 60 s, it can drive 33 light-emitting diodes for more than 45 s (Fig. 4f). The cycle stability of the device was tested at a current density of 1.2 mA cm$^{-2}$. The capacitance retention is 92% after 10,000 cycles, reflecting the excellent cycle stability of LEID (Fig. 4g).

The self-discharge performance was also measured (Fig. 4h). After being charged to 2 V, the decreasing curve of the open-circuit voltage of LEID-3 was recorded. After 50,000 seconds, the voltage dropped to 0.12 V, 22.4% of the original value. This self-discharge rate is similar to other supercapacitors in the literature[58–60] and demonstrates the long-term energy storage capability of LEIDs in practical applications. The self-discharge result also confirms that in LEID-3 there is no side reaction that causes obvious self-discharge. We further investigated the influence of deformation on the capacitance of LEID-3. The specific capacitance of a LEID-3 device was measured when the device was under a three-point bending test. Fig 4i shows that the specific capacitance of the device did not change with the deflection. SEM images (Supplementary Fig. 14) demonstrate that the structure of the device was intact after bending. The specific capacitance of the device also remained unchanged after 100 bending/releasing cycles (Supplementary Figs. 13, 15). The above results indicate that the interface between solid electrolyte and electrode was stable under small deformation. Therefore, in practical applications, the device can withstand small elastic deformation without irreversible degradation of electrochemical performance. The above results show that LEID-3 has both the high strength and electrochemical energy storage capacity and thus is a high-performance integrated device.

## Design, fabrication, and performance of multilayer load-bearing/energy-storage integrated device

From the perspective of composite materials, the thickness and absolute strength of relatively thin LEID-3 may not meet the needs of practical applications. Therefore, multilayered LEID with more CF and GF layers is necessary. Here we designed and compared two kinds of multilayered LEID. The alternate stacked CF and GF were directly used in the first design (Structure (I)). Figure 5a shows the structure of a seven-layered LEID-7(I) with a configuration of (CF/GF/CF/GF/CF/GF/CF)$_{EP50}$. The preparation method of this multilayer device is the same

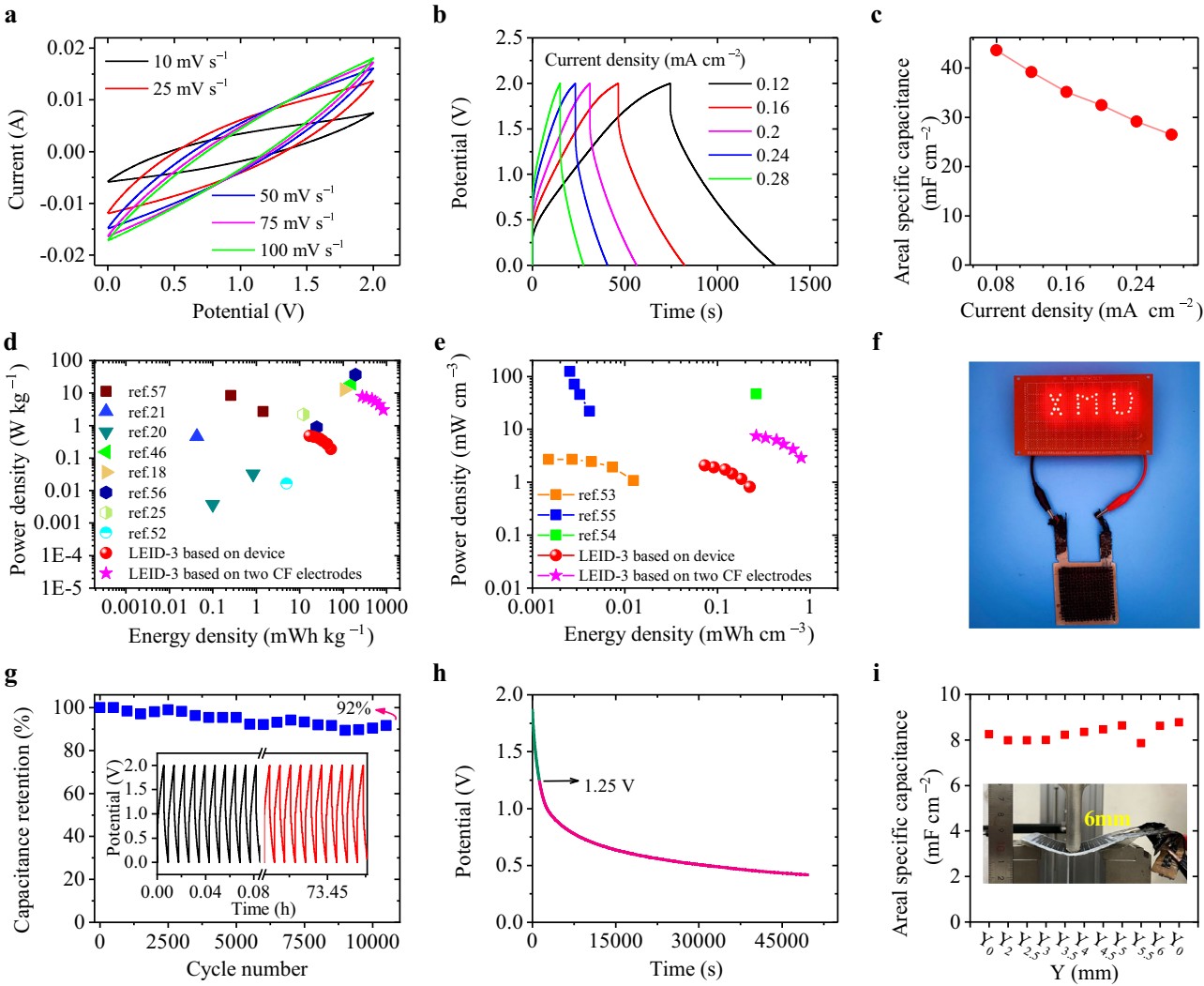

**Fig. 4 | Electrochemical properties of LEID-3. a** CV curves of LEID-3 at different scanning rates. **b** GCD curves of LEID-3 under different current densities. **c** Areal specific capacitance of LEID-3 at different current densities. **d, e** Ragone plots of the LEID-3 and literature data. (Based on device: the energy and power densities were calculated based on the mass or volume of the whole device, including the CF electrodes, GF separator, and solid electrolyte. Based on CF electrodes: the energy and power densities were calculated based on the mass or volume of the two CF electrodes or total mass of active materials and CF.) **f** The photo of a LEID-3 lighting up these LED lights. **g** Cyclic stability of LEID-3 in 10,000 cycles at a current density of 1.2 mA cm$^{-2}$. Inset shows the GCD curves of different cycles. **h** Self-discharge curve of LEID-3. **i** Areal specific capacitance of LEID-3 at different bending deflections. Inset shows the Photo of three-point bending test.

as the three-layered LEID-3, so it is technically simple. The second type of multilayered device we designed is comprised of individual three-layer LEIDs bonded by insulating layers (Structure (II)). As shown in Fig. 5b, two individual LEID-3 are bonded with a GF/EP$_{70}$ insulating layer to form a seven-layered LEID-7(II), whose structure is (CF/GF/CF)$_{EP50}$/GF$_{EP70}$/(CF/GF/CF)$_{EP50}$. EP$_{70}$ instead of pure epoxy resin was used for the insulating layer because the curing temperature of pure epoxy resin is too high and inconsistent with the curing conditions of other layers. EP$_{70}$ has a conductivity of less than 10$^{-5}$ mS cm$^{-1}$, and can be treated as an ion insulating layer in the device.

The photos of three-layer, five-layer, seven-layer, and nine-layer LEIDs with Structure (I) are shown in Supplementary Fig. 17. As the number of layers increases, the thickness of the device increases from 1.2 mm to 2.9 mm. Meanwhile, the moduli of these LEIDs are similar, and consequently, the load-bearing capacity of LEID increased with the number of layers (Supplementary Fig. 18). Taking the LEID-7(I) as an example, we can measure the electrochemical properties of three-layered and five-layered subdevices and the seven-layered whole device. Unfortunately, we found that the specific capacitance decays significantly with the number of layers. The specific capacitances of

three-, five-, and seven- layered LEID at a current density of 8 × 10$^{-2}$ mA cm$^{-2}$ were 5.1, 2.2, and 0.6 F m$^{-2}$, respectively (Fig. 5 b–d), and the rate performance of the device gradually decreases (Supplementary Figs. 19–21). One important reason for the attenuated performance is the large resistance. It can be seen that the equivalent series resistance (ESR) of the three-, five-, and seven-layered LEID are 13.3, 24.1, 49.3 Ω, respectively (Fig. 5d). In fact, although LEID-7(I) has an alternative array of CF and GF, it is not a series of three individual LEID-3. In this device structure, only the upper and lower layers of CF fabrics function as electrodes, while the other CF fabrics are just porous conductors filled with the electrolyte. When the electrostatic equilibrium is reached, the solid electrolyte is an equipotential body, so the CFs inside the solid electrolyte make no actual contribution to energy storage. Meanwhile, due to the increase in the thickness of the device and the decrease in the effective transport area caused by additional CF and GF, the resistance of the electrolyte increases significantly, resulting in an excessive voltage drop of the device. Moreover, the working voltage of the LEID-7(I) also did not increase. Therefore, multilayered LEIDs with Structure (I) cannot work in this series mode.

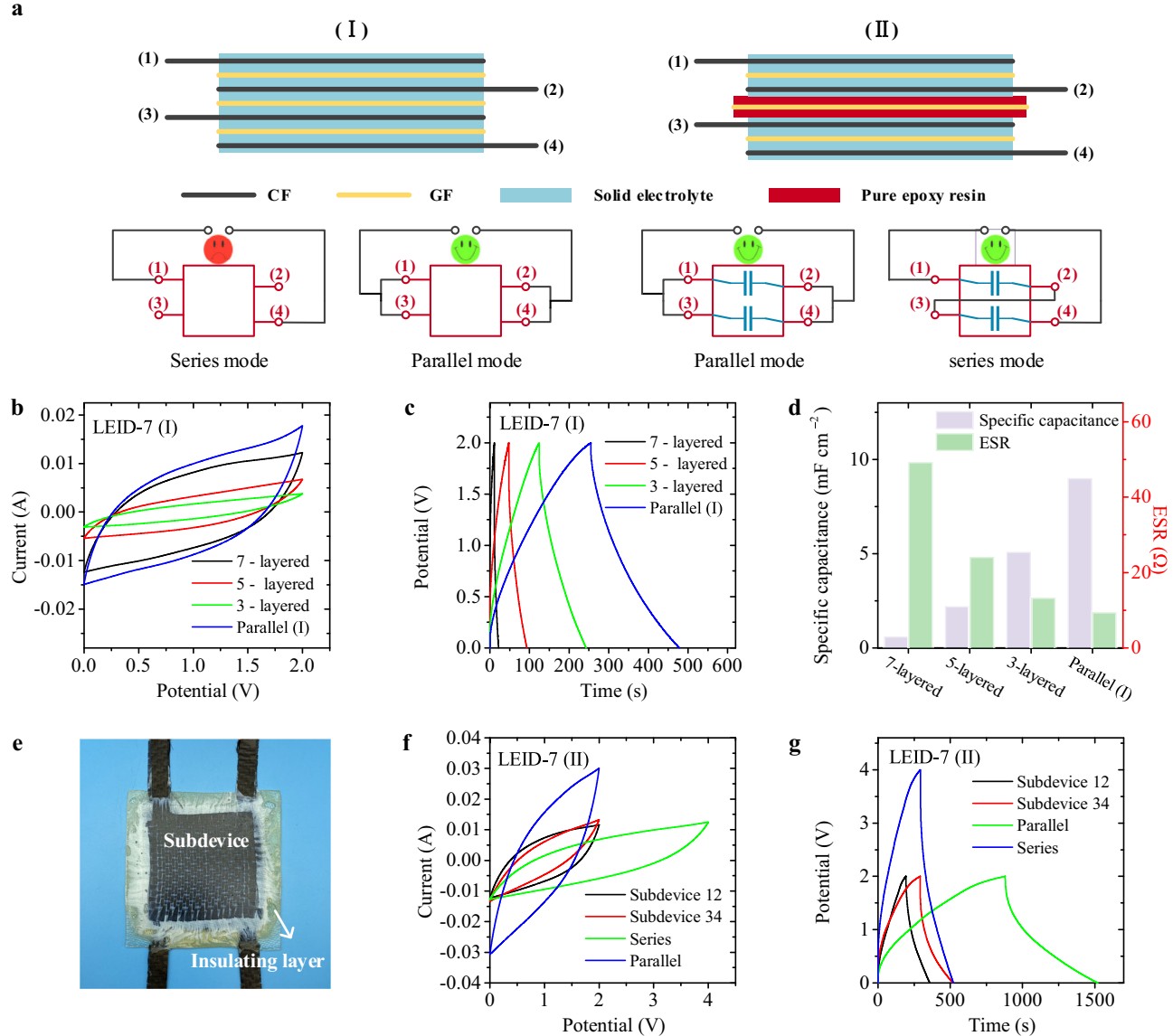

**Fig. 5 | Design and performance of multilayered LEIDs. a** Schematic diagram of two different structures and the corresponding usage methods of LEID-7. **b, c** CV curves (**b**) and GCD curves (**c**) of LEID-7(I) in parallel mode and the three-, five-, and seven- layered subdevices of LEID-7(I). **d** Area specific capacitance and ESR of three-, five-, and seven- layered subdevices of LEID-7(I) and parallelly connected LEID-7(I). Current density: 0.08 mA cm$^{-2}$. **e** Photo of a LEID-7(II). **f, g** CV curves (**f**) and GCD curves (**g**) of LEID-7(II) as a whole in series and parallel modes and the two subdevices of LEID-7(II).

However, multilayered LEID of Structure (I) can work in a parallel mode. As illustrated in Fig. 5a, CF 1 and 3 were connected to form one electrode, while CF 2 and 4 form another. LEID in this working mode is like the interdigital electrode-type planar device, where all the figures of one interdigital electrode are connected in parallel. In parallel mode, the ESR will not increase with the layer number of the CF, because the distance between the two adjacent opposite CF electrodes always keeps the same. We examined the electrochemical properties of parallel-connected LEID-7(I). The CV curves show that at the same scanning rate, parallel-mode LEID-7(I) had a larger charging current than three-layered, five-layered subdevice and series-mode LEID-7(I). The specific capacitance of parallel-mode LEID-7(I) at a current density of $8 \times 10^{-2}$ mA cm$^{-2}$ was 9.0 mF cm$^{-2}$ (Supplementary Fig. 22), larger than that of the three-layered subdevice. Meanwhile, the ESR of parallel-mode LEID-7(I) was only 9.4 Ω, smaller than the three-layered subdevice. These results can be well explained by the parallel connection of capacitors. Although the parallel-connected LEID-7(I) has a working voltage of 2.0 V, which is the same as the three-layered subdevice, its

areal energy density was 43.3% higher than that of the three-layered subdevice due to the higher areal specific capacitance.

The Structure (II) is another option for multilayered LEID. In this structure, the two LEID-3 devices formed a four-terminal LEID-7(II). We first used interlaminar shear strength to examine the bonding strength of the insulating layer. Figure 5e shows the photo of a LEID-7(II). The interlaminar shear strength of LEID-7(II) is slightly larger than that of LEID-3 (14.5 MPa *vs.* 13.5 MPa, Supplementary Fig. 23), showing that the two LEID-3 devices are well bonded by the EP$_{70}$ insulating layer. The polymer matrix of both the LEID-3 and insulating layer are the same, so they have excellent interface bonding. Since the two LEID-3 subdevices are independent, they can work in both series or parallel modes. Figure 5f, g shows the CV and GCD curves of LEID-7(II). When the current density is $8 \times 10^{-2}$ mA cm$^{-2}$, the specific capacitances of the two LEID-3's are 6.9 mF cm$^{-2}$ and 7.0 mF cm$^{-2}$, respectively. After bonding, the areal specific capacitance of LEID-3 was unchanged, showing that the EP$_{70}$ layer has no significant effect on the two subdevices. CV and GCD curves show that the operating voltage range of a series-mode LEID-7(II) can be up to twice that of LEID-3, but the areal specific capacitance

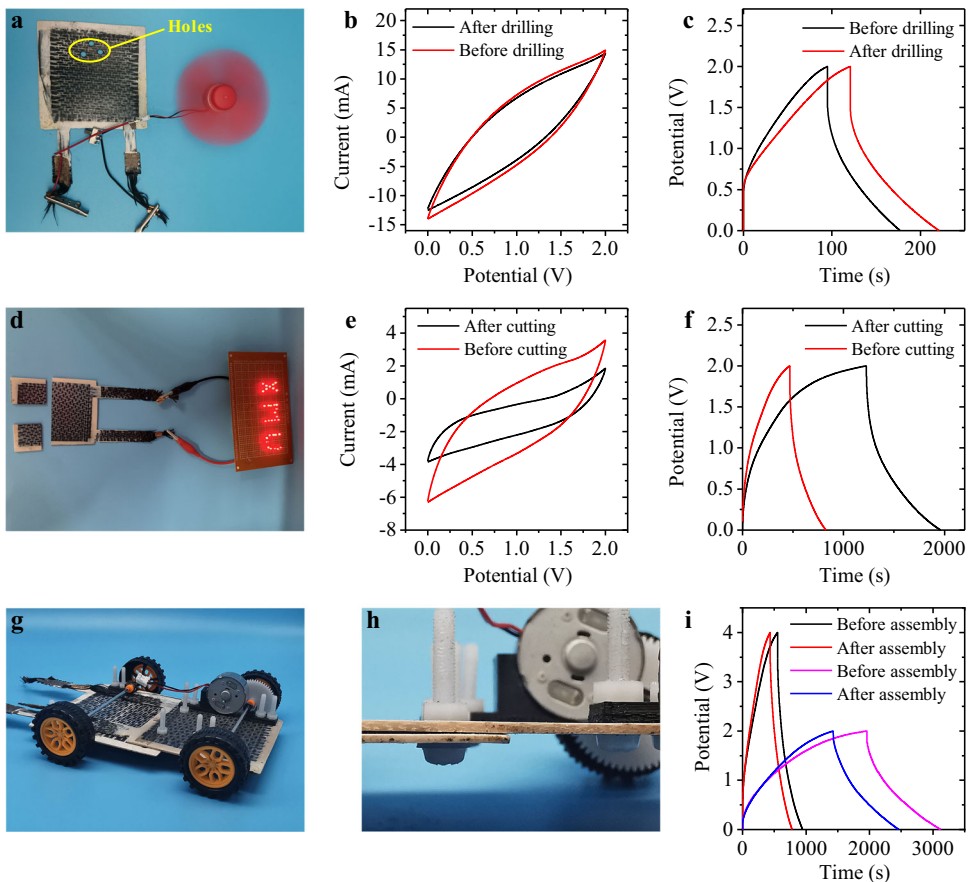

**Fig. 6 | Secondary processing of LEIDs. a** The photo of a drilled LEID-3 as a power source to drive a fan. **b**, **c** CV curves (**b**) and GCD curves (**c**) at current density of 0.5 mA cm⁻² of LEID-3 before and after drilling. **d** The photo of a LEID-3 lighting up the LEDs after being cut into thrr pieces. **e**, **f** CV curves (50 mV s⁻¹) (**e**) and GCD (**f**) curves (1.8 × 10⁻² mA cm⁻²) of LEID-3 before and after being cut. **g**, **h** The photos of a car model with two LEID-3 plates assembled by plastic fasteners as both the chassis and the power supply. **i** GCD curves of the assembled LEID-3 chassis (0.2 mA cm⁻²).

drops to about half of LEID-3. When tested in parallel mode, LEID-7(II) showed a capacitance twice that of LEID-3. The specific capacitance LEID-7(II) was measured to be 4.2 mF cm⁻² in series mode and 21.6 mF cm⁻² in parallel mode (Supplementary Fig. 24). Therefore, although the fabrication of the Structure (II) device is more complicated than Structure (I), the insulating layer provides us with a flexible way of using the devices. One can choose to use the device in series or parallel modes to obtain a higher working voltage or a larger capacitance to meet the needs of different applications.

### Secondary processing of load bearing-energy storage integrated devices

The effect of structural damage on the energy storage is critical for LEID because the structural materials used for load-bearing are at the risk of fracture caused by external forces. Here we evaluate the energy storage retention of our LEID under damages[9]. Figure 6a–c shows that after three holes were drilled on a LEID-3, the CV and GCD curves of the device did not change, indicating that it is not short-circuited and can still work normally. The drilled LEID-3 could still power a fan (Fig. 6a and Supplementary Movie 2). We also cut a LEID-3 into two pieces, as shown in Figure 6d–f, and tested the electrochemical performance. The CV and GCD tests show that the device maintained a certain capacitance after cutting, and no short circuit or other failure was observed. When the current density is 1.8 × 10⁻² mA cm⁻², the specific capacity is 6.9 mF cm⁻² before cutting and 6.8 mF cm⁻² after cutting. The total capacitance of the device decreased after cutting, but the areal capacitance was similar. Fig. 6d shows that the half LEID-3 can also light up 33 LEDs. Actually, because the solid electrolyte has high

mechanical strength, it can efficiently separate the CF electrodes during the perforation and cutting and thus prevent the internal short circuit. The above results show that the LEIDs can still work when subjected to structural damage, so these devices have high safety.

Because LEID can be cut and drilled without performance degradation, we can perform secondary processing on the LEIDs and assemble them into complex structures with other parts. Figure 6g, h show that two LEID-3's were drilled and assembled with plastic fasteners to form a larger plate. After connecting two LEID-3's in series, the newly assembled sheet works well as an electrochemical capacitor, as indicated by the GCD curves in Fig. 6i. This sheet was further used as the chassis to make a car model with an electric motor. All the parts in this model were assembled with plastic fasteners. With LEID-3 as both the chassis and power supply, the car model can run for a long distance (Supplementary Movie 3). This example demonstrates that our LEID can be conveniently assembled and used like other conventional structural materials. This ability is important for the practical application of LEID.

## Discussion

Through the in-situ cross-linking induced phase separation between the 5284 epoxy resin and the ionic liquid, a bi-continuous phase structure polymer solid electrolyte with a high modulus (flexural modulus -1.0 GPa) and high ionic conductivity (6.7 × 10⁻¹ mS cm⁻¹) was obtained. Using this solid electrolyte and CF electrodes, we built high-performance LEIDs, whose areal specific capacitance was 32.4 mF cm⁻² at the current density of 0.2 mA cm⁻², and flexural modulus and flexural strength were 18.1 GPa and 160.0 MPa, respectively, higher than

those of common engineering plastics. As new designs, two basic structures of multilayered LEID were devised and compared. The multilayered device with Structure (I) had a simpler structure and was easier to prepare, but it could only work in parallel mode as an electrochemical capacitor, because the inner CF electrodes made no contribution to the energy storage. The device with Structure (II) with a separating layer can work in both series and parallel modes, although the preparation is more complicated. We also demonstrated that the LEIDs could be processed into complex shapes composed of curved and flat surfaces, and secondarily machined and assembled into a complex system. This work showed the potential applications of LEID, and we believe that with a rational design, the LEIDs can provide new feasible solutions for increasing the energy density of mobile systems.

## Methods

### Materials

All chemical reagents were purchased from commercial sources and used without further purification. 5284 epoxy resin (75 wt% E54, 20 wt% cyanate resin 5 wt% titanium acetylacetonate) was provided by AVIC Composite Corporation Ltd, 1-Ethyl-3-methylimidazolium bis(trifluoromethanesulfonyl)imide (EMIM-TFSI) was purchased from Lanzhou Yulu Fine Chemical Co., Ltd. Bis(trifluoromethane) sulfonamide lithium salt (LiTFSI) and propylene carbonate (analytical pue, AR) were supplied by Aladdin. Carbon fiber (CF-GW303) was purchased from Shandong Weihai Development Fiber Co., Ltd. The CF fabric is a satin fabric with an areal density of $220 \pm 7 \, g \, cm^{-2}$ and a thickness of $0.250 \pm 0.025 \, mm$. The diameter of a single CF fabric is $10 \pm 0.5 \, \mu m$. The specific surface area was measured to be $0.44 \, m^2 \, g^{-1}$, and the average pore size was $5.3 \, nm$ (Supplementary Fig. 16). It is worth mentioning that the carbon fiber is not modified with any active material. The GF (SW280F-90a) was bought from Nanjing Fiberglass Research and Design Institute Co., Ltd. The areal density of GF is $280 \pm 20 \, g \, cm^{-2}$, and the thickness of GF is $0.250 \pm 0.025 \, mm$, the diameter of a single glass fiber is $8 \pm 0.5 \, \mu m$. Acetone (AR) and Ethanol (AR) were supplied by Sinopharm Chemical Reagent Co., Ltd. Copper foil was purchased from Guangzhou Lige Technology Co., Ltd.

### Preparation of solid electrolyte

$11.8 \, g$ EMIM-TFSI, $0.1 \, g$ PC, and $5.2 \, g$ LiTFSI were mixed in a nitrogen-filled glove box by stirring for $12 \, h$ to obtain a uniform LE. To prepare the precursor of $EP_{50}$, $2.8 \, g$ LE and $2 \, g$ of epoxy resin prepolymer were mixed by stirring and then degassed in a vacuum oven at $60 \, °C$ for $15 \, min$, until a uniform and clear solid electrolyte precursor mixture was obtained. Other solid electrolyte precursors were obtained by changing the feeding mass ratio of EP and LE.

The solid electrolyte precursor mixture was pre-cured at $100 \, °C$ for $8 \, min$ to increase the viscosity to avoid leakage during in following procedures, and then transferred into the mold and cured using temperature programming. The temperature program is $80 \, °C$ for $1 \, h$, $100 \, °C$ for $1 \, h$, and $120 \, °C$ for $3 \, h$. After cooling down to room temperature, a solid electrolyte casting body was obtained from the mold. To prepare pure epoxy resin, the EP prepolymer was cured at $160 \, °C$ for $1 \, h$, $170 \, °C$ for $1 \, h$, and $180 \, °C$ for $3 \, h$.

### Fabrication of LEIDs

The CF and GF were washed three times with ultrapure water, soaked in ethanol for $30 \, min$, and then dried in vacuum at $40 \, °C$ for $6 \, h$. The CF was cut into a square of $55 \, mm \times 55 \, mm$ with a $60 \, mm \times 8 \, mm$ strip as one side as a lead, and the GF was cut into a square of $65 \, mm \times 65 \, mm$. The solid electrolyte precursor mixture is evenly coated on CF and GF and fully infiltrated to prepare prepregs[61]. The CF and GF prepregs were laid alternatively into a mold, as shown in Fig. 2a, and cured by hot-pressing. The pressure was $2 \, MPa$, and the temperature program was the same as that of the solid electrolyte. After curing and demolding, a LEID was obtained.

When preparing a multilayer LEID with Structure (II), two independent LEID-3 were bonded by an insulating layer of a GF pre-impregnated with $EP_{70}$ precursor. The device was hot-pressed following the same procedure as described above to cure the insulating layer.

### Electrochemical tests

All the electrochemical measurements were performed on a 660D electrochemical workstation (CHI, USA) with a two-electrode mode. The electrodes in the electrochemical test is bear CF electrode without any active material, so the mass loading equals to the areal density of the CF ($220 \pm 7 \, g \, cm^{-2}$). No pre-activation of the electrode for the electrochemical testing was performed. Cyclic voltammetry (CV) and galvanostatic charge/discharge (GCD) test voltage range is $0-2 \, V$. The voltage range of the devices connected in series is $0-4 \, V$. The test range of cathode and anode current density is $0.08-1.2 \, mA \, cm^{-2}$. All the electrochemical tests were carried out under the atmosphere and at $25 \, °C$. The specific capacitances of LEID were calculated according to GCD curves following Eq. (1):

$$C_D = \frac{It}{(U - IR)K_D} \tag{1}$$

where $I$ is the current applied to the device, $t$ is the discharge time, $U$ is the maximum voltage of the GCD process, $IR$ represents the voltage drop at the beginning of the discharge. When $K_D$ is the area, mass, or volume of the device (the mass and volume of the separator GF and solid electrolyte are included), the result of (1) is areal, gravimetric, or volumetric specific capacitance, respectively. The specific capacitance of one electrode was calculated by:

$$C_E = \frac{2K_D}{K_E}C_D \tag{2}$$

where $K_E$ is the area, mass, or volume of one CF fabric electrode.

The energy density ($E_D$) and power density ($P_D$) of the device were calculated by following Eqs. (3) and (4):

$$E_D = \frac{1}{2}C_D(U - IR)^2 \tag{3}$$

$$P_D = \frac{E_D}{t} \tag{4}$$

where $C_D$, $t$, $U$, and $IR$ are the same as defined in Eq. (1). The energy density ($E_E$) and power density ($P_E$) of one electrode were calculated by:

$$E_E = \frac{K_D}{2K_E}E_D \tag{5}$$

$$P_E = \frac{E_E}{t} \tag{6}$$

Where $E_D$, $K_D$, $K_E$, and $t$ are the same as defined above.

To measure the self-discharge curve, the device was charged to $2 \, V$ at a current density of $0.16 \, mA \, cm^{-2}$ and then disconnected from the test circuit to rest for $20 \, min$, before the open circuit voltage change was begun to record.

### Characterization

The morphology of the composites was observed using a SU-70 field emission scanning electron microscope (Hitachi, Japan) operated at $10 \, kV$. Tensile tests were performed on an AGS-X electronic universal testing machine (Shimadzu, Japan) with a speed of $10 \, mm \, min^{-1}$ and

the typical specimen dimension of 80 mm × 10 mm × 4 mm. The bending performance of the material was tested with the AGS-X electronic universal testing machine with a specimen size of 20 mm × 6 mm × 2 mm. Fourier transform infrared spectra were obtained on an iS10 spectrometer (Thermo Fisher, USA). Thermogravimetric analysis under nitrogen atmosphere was carried out on an STA 409EP Simultaneous Thermal Analyzer (Netzsch, Germany) from 30 °C to 800 °C at a heating rate of 10 °C min$^{-1}$. DSC data were collected on a 204 F1 differential scanning colorimetry (Netzsch, Germany) from 25 °C to 350 °C at a heating rate of 10 °C min$^{-1}$ under nitrogen atmosphere. Dynamic thermomechanical analysis was conducted with a 242E Analyzer (Netzsch, Germany) from 25 °C to 250 °C with a heating rate of 3 °C min$^{-1}$. The tests were performed in the double cantilever beam mode with a sample size of 60 mm × 10 mm × 4 mm, a frequency of 1.0 Hz and an amplitude of 5 μm. Gas adsorption tests were carried out on a ASAP 2460 2.02 (MicroActive, USA). The degassing temperature was 350 °C, and the degassing time was 8 h.

## Data availability

The data that support the findings and conclusions of this study are available in the Article and its Supplementary Information file. Additional information with relevance is available from the corresponding authors on reasonable request.

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

## Acknowledgements

H.B. acknowledges the financial support from the Natural Science Foundation of China (21975210, 22179115), and the Youth Innovation Fund of Xiamen (3502Z20206043). Thanks to Mr. Changjian Chen from Xiamen University for his help in instrument testing.

## Author contributions

H.B. conceived the concept, H.B. and X.L.H. designed the experiments, supervised the research, and wrote the paper. J.M.Z. and J.L.Y. performed the experiments, collected and analyzed the data, and wrote the paper. Y.N.Z. collected SEM images. Q.Z. designed and assembled the model car. Y.X.M. helped the electrochemical measurements. Y.X.Z. analyzed mechanical properties. A.A.Z., L.H.L., and S.M.L. analyzed the experimental data. All authors discussed the results and commented on the manuscript.

## Competing interests

All authors declare no competing interests.
