## [Peer Review File · Nature Communications]

High-strength and machinable load-bearing integrated electrochemical capacitors based on polymeric solid electrolyteREVIEWER COMMENTS

Reviewer #1 (Remarks to the Author):

The authors present a very original work, based on a technology called LEID, Load bearing/energy Storage Integrated devices. The mechanical properties are impressive, combined with a conservation of conductivity properties. However, a number of elements are missing before we can conclude on the true interest of the work. There is no doubt that the mechanical properties are solid (proven by some very interesting supporting data), the missing element concerns rather the electrochemical measurements. These will be described in the next section. The comments are separated into two parts: major comments (which limit the actual interest of the work) and minor comments.

Major comment: lack of comparison with the State-Of-The-Art. The most severe criticism of the present work is the lack of comparison with the State-Of-The-Art, (I) the chemical nature of the products used in the present paper, (II) the electrochemical performances, and (III) the possible applications.

(I) The chemical nature of the products used.

(I.1) P.3 L.35. The authors mention the problem of dead mass. This point is dealt with too superficially. A table in supporting information, with a comparison of the materials used in the LEID, presenting different mass properties should be discussed in order to situate the present study. We do not have any mass normalisation to conclude whether the present materials offer a lower dead mass. This is very limiting for assessing the quality of the scientific paper. Especially since this element is presented as a strength of the work.

(I.2) P.3 L.45. The authors mention the LEIDS materials. However, here again it is too superficial and it is difficult to locate the advantage and limitation of the approach proposed by the authors. Again, a table, or a figure with supporting information is needed to situate the present study. In particular, here it would be necessary to have performance values with comparable metrics (by mass, by area, and by volume). A Ragon-type diagram would also be appreciated. This last point will be discussed in criticism (ii).

(I.3) P.5 Fig.1 (or P.6 L 82). The authors propose an interesting solid-state electrolyte. However, the chemical nature of the reagents is not sufficiently justified. In particular the nature of the ionic liquids, why not use a more conventional electrolyte? The comparison with other salts would be an interesting way (however, I do not ask the authors to carry out the measurements, because it requires too much work). In the absence of additional measurements, it would be necessary to justify the choice of ionic liquid with State-Of-The-Art support. Concerning the epoxy precursor, it also needs more justification, why not a simple PEG? Without these elements of justification, the article is weakened, especially with regard to the proposed generic journal.

(I.4) P.8 L 132- 145. A more serious comparison is needed, measuring the ionic conduction of ionic liquid, LE. As well as a comparison with the State-of-The-Art. Again, a table in the supporting information would give credibility to the study.

(II) Electrochemical measurements. Important weaknesses are identified in the present form of the manuscript. Here the list, without the metrics asked, the results are not enough strong.

(II.1) The authors should definitely perform a Ragon diagram in to position the present work with the State-Of-The-Art. This is essential.

(II.2) The authors must, without negotiation, make a self-discharge measurement, as self-discharge is a limiting parameter for capacitor technology.

(II.3) Authors should perform a stability measurement over more cycles (minimum 10,000 cycles) currently, they show 5,000 cycles.

(II.4) The authors make measurements within a cell-voltage limit close to 2 Volts, why? In view of the salts used, a voltage of 3-4 Volts should be accessible. A measurement of the maximum voltage would then be necessary to realize the extent of the interest of the proposed materials. Note that the voltage impacts both the energy density and the max power. Without a more rigorous measurement of this cell voltage, the article loses a major interest. Moreover, on P.14 Fig. 4, the authors show figure b, c at 2 volts and figure f and g at 4 volts. This is not well justified.

(II.5) The authors normalize their measurements per area, whereas normalization per mass and volume would be necessary.

(III) The present work, based on a LEID technology, is interesting, however, it would be appreciated if in the introduction the current or envisaged applications were a little more detailed. Nature Communication is a generalist journal with a wide readership.

(IV) Minor remarks:

(IV.1) P.1 L.15. The description of the state of the art in the introduction is too short, a sentence on the importance of technology is needed.

(IV.2) P.1 L. 17. The authors write « high-performance », this term is too empty, it is necessary to be more precise, values should be given.

(IV.3) P.2 L 27. The authors should mention the values of energy density and maximum power obtained.

(IV.4) P.4 L 61. A comparison with the state-of-the-art on solid-state-electrolyte would be appreciated.

(IV.5) P.9 L 153. In the title the authors note LEID-3. This should be avoided, especially the 3, which refers to an internal notation and not an internationally accepted nomenclature.

(IV.6) SI P.2 L18. The composition in the table is not sufficiently explained.

Reviewer #2 (Remarks to the Author):

Attached were Reviewer's comments

Reviewer #2 (Remarks to the Author):

371400 Zhang, et al., “Machinable multilayered load-bearing/energy storage integrated electrochemical capacitor based on high-performance polymeric solid electrolyte”.

This paper presents the polymeric solid electrolytes with both mechanical strength and ionic conductivity for the development of load bearing/energy storage integrated devices (LEIDs). The authors prepared a series of epoxy resin-based solid electrolytes containing LiTFSI dissolved in EMIm-TFSI ionic liquid via the thermal curing process, and investigated their ionic conductivity and mechanical properties, combined with the morphological study. Then, the authors designed the multilayered LEIDs based on the electrolyte, CF, and GF, and explored their electrochemical performance under mechanical and machinable deformation. These results could be useful to the readers who are interested in the development of multifunctional structural devices with both mechanical bearing capacity and electrochemical storage capacity, but there still exist questionable points that need to be addressed before this paper can be published, detailed below.

1. Page 5, Fig. 1a

: In Fig. 1a, “LiTSFI” should be replaced with “LiTFSI”.

2. Page 5, Lines 95-96 & Page 21, Lines 371-375

: To determine the curing temperature, the authors conducted the DSC measurement (SI, Fig. 1). The authors also mentioned that “As the content of ionic liquid (IL) increases, the exothermic peak shifts to lower temperature.”. However, the peak temperature seems to have a nonmonotonic IL concentration dependence; i.e., the EP50 with the lower IL content has a lower peak temperature (117 C) than the EP40 with the higher IL content (123 C). Can the authors explain such a nonmonotonic dependence? In addition, how about the correlation between the IL content and the enthalpy of the exotherms (ΔH)? In the section of “Preparation of solid electrolyte”, the authors mentioned that the solid electrolyte was pre-cured at 100 C for 8 min. Is there any specific reason for performing the pre-curing process at 100 C? Although the exothermic peak of the EP prepolymer was observed at 275 C from the DSC curve (SI, Fig 1), it was cured at 160, 170, and 180 C, not at ~275 C. Can the authors also explain this point?

3. Page 7, Lines 104-107 & Lines 116-117

: The authors mentioned that “the solid electrolytes have a bi-continuous structure composed of a LE phase and a LE-plasticized epoxy resin phase.”. If the solid electrolytes have the bi-continuous structure, then the bi-continuous electrolytes would be expected to have two glass transition temperatures (T_g s). However, from the DMA measurement (SI, Fig 5), it looks like the electrolytes have a single T_g , which is presumably attributed to the LE-plasticized epoxy phase. What about the T_g from the LE phase of the solid electrolyte? DSC measurements may be useful to observe the LE-phase T_g at lower temperatures.

4. Page 8, Lines 132-133

: The authors mentioned that “the ionic conductivity of the solid electrolyte increased with the LE” (Fig. 1f, g and SI, Fig. 7). However, in SI Fig. 7 (Nyquist plot), the EP45 seems to have lower resistance than the EP40, which is not consistent with the result shown in Fig. 1g. The authors should re-check this conductivity result.

5. Page 8, Lines 143-145

: The references cited in the sentence seem to be different from those in Fig. 1h. If so, the authors should ensure that all the references are correctly cited in this manuscript.

6. Page 10, Line 157

: What is GF? Is it glass fiber? There is no information about GF, such as thickness, pore size, etc., in this manuscript. For the LEID fabrication, the authors seemed to use GF as the separators. Even though the solid electrolyte itself can act as a separator, why did the authors use GF?

7. Page 10, Lines 163-164, SI Fig 9b

: For the electrochemical performance of the LEID-3 car shell in SI Fig. 9b, the GCD curve at the current density of 0.4 A/cm^2 is missing. Please revise the figure.

8. Page 12, Line 202 & Page 17, Line 303

: There are several typos in the numbers of the main and SI figures; for example, “Fig. 3d” should be replaced with “Fig. 3c” and “Supplementary Fig. 16” should be replaced with “Supplementary Fig. 17”. The authors should re-check all the figure numbers.

9. Page 13, Lines 213-214 & Fig 3e

: The authors mentioned that “the current densities of the CV curves are proportional to the scanning rate showing a typical capacitive behavior”. The authors can directly prove it by plotting current vs. scan rate and by analyzing the slope of the linear dependence.

10. Page 13, Lines 216-218

: The authors indicated the calculated areal specific capacitance, energy density, and power density of LEID. The comparison of this device with the state of the art should also be presented in Ragone plot and discussed in the text.

11. Page 18, Lines 324-326 & Fig. 5i

: The authors mentioned that after the secondary processing, the new assembled sheet works well as an electrochemical capacitor, as indicated by the GCD curves in Fig. 5i. However, the GCD curves in Fig. 5i exhibit considerable IR drop, unlike the GCD curves shown in Fig. 4c. The authors should directly compare and explain the GCD curves before and after the secondary processing.

12. Page 18-19 & Fig. 5

: For the stable and rigid external impact resistive device, the higher mechanical strength and flexibility should be given. From the images of a car model shell made of LEID-3 (Fig. 2c) and of different curve shapes (SI, Fig. 8), the device seems to have flexibility. However, there is no investigation for electrochemical performance under mechanical deformation such as capacitance retention as a function of bending radius or bending cycle numbers. Furthermore, bending can cause the local expansion/contraction of the electrodes and electrolyte. This will induce interface contact failure, especially with solid-state devices. As a result, poor contact between electrodes and electrolyte is a critical issue for solid-state electrochemical devices. Therefore, the authors should provide the cross-section image of electrode-electrolyte integration before and after the mechanical deformation. This will further demonstrate that the LEID has high electrochemical stability.

13. Page 21

: For the CF electrode used in the device, its specific surface area, mass loading, or size is not given in this manuscript. The authors should provide such information about the electrode used for capacitor evaluation.

Point-by-Point Response to Reviewers' Comments

Review #1

The authors present a very original work, based on a technology called LEID, Load bearing/energy Storage Integrated devices. The mechanical properties are impressive, combined with a conservation of conductivity properties. However, a number of elements are missing before we can conclude on the true interest of the work. There is no doubt that the mechanical properties are solid (proven by some very interesting supporting data), the missing element concerns rather the electrochemical measurements. These will be described in the next section. The comments are separated into two parts: major comments (which limit the actual interest of the work) and minor comments.

Response:

Thank you very much for your constructive comments on our manuscript. We have revised the manuscript following your suggestions, and the revision we made are described below.

Comment 1

Major comment: lack of comparison with the State-Of-The-Art. The most severe criticism of the present work is the lack of comparison with the State-Of-The-Art, (I) the chemical nature of the products used in the present paper, (II) the electrochemical performances, and (III) the possible applications. (I) The chemical nature of the products used.

(I.1) P.3 L.35. The authors mention the problem of dead mass. This point is dealt with too superficially. A table in Supplementary information, with a comparison of the materials used in the LEID, presenting different mass properties should be discussed in order to situate the present study. We do not have any mass normalisation to conclude whether the present materials offer a lower dead mass. This is very limiting for assessing the quality of the scientific paper. Especially since this element is presented as a strength of the work.

Response:

Thank you for your valuable suggestions. The discussion of the dead mass in previous manuscripts is not very clear. The concept of “dead mass” in our manuscript is from the energy storage perspective

and based on the whole system, and we did not discuss the internal dead mass of the LEID or battery. The energy density of a system should be defined as the ratio of the total energy stored in the system (usually in the batteries) to the system's total mass. Therefore, all the parts in the system that do not store energy should be counted as the dead mass because they only add to the denominator. For example, the car shell and chassis are dead mass in an electric vehicle because their energy density is zero. These parts usually do not have the function of energy storage, but they occupy a large proportion of the mass and volume of the whole system. The weight of the car shell and chassis can account for 40% of the weight of the whole vehicle. If the metal car shell and chassis can be replaced by the LEID, the energy density of the electric vehicle will increase because the LEID car shell and chassis are not dead mass: they can store energy. Moreover, the vehicle's total mass will be reduced because of the lower density of LEID (0.17 g cm^{-3}) compared with metal (1.81 g cm^{-3}) for magnesium-aluminum alloy, the light material for sport-car shell). Therefore, using LEID to replace some structural materials will reduce the dead mass of the system.

The internal dead mass of an electrochemical energy storage device is a different concept. The internal dead mass of a battery or a supercapacitor includes the shell, current collector, separator, and other auxiliary parts. In our lab, in a 2032 coin lithium-ion battery with lithium foil as the anode and LiFePO_4 as the cathode, the dead mass can be up to $\sim 95\%$ (**Table R1**). While for our LEID, the dead mass was $\sim 38\%$, because LEID does not have the shell and other auxiliaries (**Table R2**).

We rewrote the corresponding part in the introduction to make it more straightforward:

“From the perspective of energy storage, load-bearing components in conventional power supply systems can be defined as dead mass⁷, which reduces the total energy density of the system by increasing the denominator^{4, 6, 12, 13}. For example, in an electric car, the metal shell and chassis are dead mass because they make up 40% of the car's weight, but they have zero energy density⁷. If the energy-storage component has sufficient strength and can serve as mechanical support, it can replace the structural component. For the whole system, the total energy density is increased because the usage of dead mass can be reduced^{14, 15}.” (Page 3, Line 30 ~ 36)

Table R1. The quality of the individual components of the lithium-ion battery.

Battery parts	m / mg
Lithium foil	93.7
LiFePO ₄	1.0
Current collector	0.6
2032 positive shell	888.5
2032 negative shell	855.7
Gasket	3590.1
Separator	3.2
1 M LiPF ₆ in EC : DMC =1 : 1 vol%	131.4

Table R2. The quality of individual components of solid-state supercapacitors.

LEID parts	m / mg
Current collector (CF)	1898.0
Separator (GF)	1780.0
Solid electrolyte	6069.0

Comment 2

(I.2) P.3 L.45. The authors mention the LEIDS materials. However, here again it is too superficial and it is difficult to locate the advantage and limitation of the approach proposed by the authors. Again, a table, or a figure with Supplementary information is needed to situate the present study. In particular, here it would be necessary to have performance values with comparable metrics (by mass, by area, and by volume). A Ragon-type diagram would also be appreciated. This last point will be discussed in criticism (ii).

Response:

Thank you for your valuable suggestions. Following your suggestions, we compare the specific capacitance (by mass, area, and volume) (**Supplementary Table 5, Supplementary Fig. 12**) of LEID-3 with other solid devices based on similar electrode materials. The specific capacitance of LEID-3 is much higher than that of devices with CF as the electrode in the literature and comparable to some devices with electroactive electrode materials.

We also compared the energy density and power density (**Supplementary Fig. 13**) of LEID-3 with the literature data. However, there is little data on areal energy density and power density in the literature, so we mainly focused on comparing the gravimetric and volumetric energy and power density. **Supplementary Fig. 13** shows that our energy densities are higher than the literature data and are even higher than those of devices using electroactive material-modified CF as electrodes. The high energy density is ascribed to the large capacitance and high working voltage. The above data demonstrate that our LEIDs have good energy storage performance comparable with other solid devices in the literature.

A table and two figures were added to the Supplementary information, and a paragraph was added to the main text:

Supplementary Table 5. Comparison of the specific capacitance of LEID in this work with other solid-state supercapacitors in the literature.

Electrode	Separator	Electrolyte	Specific capacitance	Reference
CF	GF	EP-IL	$C_a=32.4 \text{ mF cm}^{-2b}$, $C_v=1297.8 \text{ mF cm}^{-3b}$, $C_g=675.9 \text{ mF g}^{-1b}$	This work
MnO₂-CF	GF	EP-IL	$C_a=5.68 \text{ mF cm}^{-2b}$, $C_v=82$ mF cm^{-3b} , $C_g=49 \text{ mF g}^{-1b}$	24
Vertical Graphene/MnO₂-CF	GF	EP-IL	$C_v=30.7 \text{ mF cm}^{-2b}$	19
CuO-CF	GF	Polyester -LiTf-EMIMBF₄	$C_g=6.75 \text{ F g}^{-1a}$	15
Activated CF	FP	EP-TEABF₄	$C_g=25.4 \text{ mF g}^{-1b}$	25
Carbon aerogel-CF	GF	PEGDGE-IL	$C_a=3.15 \text{ mF cm}^{-2b}$, $C_v=34.6 \text{ mF cm}^{-3b}$, $C_g=71.2 \text{ mF g}^{-1b}$	2
Graphene nanoplatelet-CF	FP	DGEBA-LiClO₄	$C_v=118.7 \text{ mF cm}^{-3b}$	23
Urea-Activated GO-CF	GF	PEGDGE-IL	$C_v=82.3 \text{ mF cm}^{-3b}$	17
ZnO-CF	GF	Polyester -LiTf-EMIMBF₄	$C_g=10.6 \text{ F g}^{-1a}$	11
Cu-Co-Se-CF	KF	Polyester -LiTf-EMIMBF₄	$C_g=28.6 \text{ F g}^{-1a}$	18
MWCNTs-CF	GF	PEG-LiTf	$C_g=125 \text{ mF g}^{-1c}$	26

^a Specific capacitance was calculated based on the mass of active materials. (without the mass of CF and electrolyte)

^b Specific capacitance was calculated based on the mass of electrodes. (The mass of CF or total mass of active materials and CF)

^c Specific capacitance was calculated based on the mass of device. (Supplementary Page 36, Line 209 ~ 213)

Supplementary Fig. 12 a-b Specific capacitance of LEID-3. Based on device: the specific capacitance was calculated based on the mass or volume of the whole device, including the CF electrodes, GF separator, and solid electrolyte. Based on GF electrodes: the specific capacitance was calculated based on the mass or volume of the two GF electrodes. (Supplementary Page 17, Line 148)

Supplementary Fig. 13 a-c Ragone plot of LEID-3 and some solid devices in the literature^{2, 11, 15, 17-23}. Based on device¹⁴: the energy and power densities were calculated based on the mass or volume of the whole device, including the CF electrodes, GF separator, and solid electrolyte. Based on GF electrodes^{11, 15, 18}: the energy and power densities were calculated based on the mass or volume of the two GF electrodes. (Supplementary Page 18, Line 153)

“According to the GCD curves, it can be calculated that at a current density of $0.2\ mA\ cm^{-2}$, the areal specific capacitance of LEID-3 is $32.4\ mF\ cm^{-2}$ (Fig. 3h, or $675.9\ mF\ g^{-1}$, $1297.8\ mF\ cm^{-3}$, see Supplementary Figure 12), surpassing those of other solid electrochemical capacitors with CF as the electrode (Supplementary Table 5, Supplementary Figure 12). ...The energy density and power density are calculated to be $0.13\ Wh\ m^{-2}$ ($3.4 \times 10^{-2}\ Wh\ kg^{-1}$, $144.9\ Wh\ cm^{-3}$) and $1.3\ W\ m^{-2}$ ($0.34\ W\ kg^{-1}$, $1453.3\ W\ cm^{-3}$). The Ragone plot (Supplementary Fig. 13) shows that the energy densities of LEID-3 are higher than other solid devices with similar structures and electrode materials in the literature. These data are even higher than those of devices using electroactive material-modified CF as electrodes⁶. The high energy density is ascribed to the large capacitance and high working voltage of LEID-3. The cycle stability of the device was tested at a current density of $1.2\ mA\ cm^{-2}$. The

capacitance retention is 92% after 10,000 cycles, reflecting the excellent cycle stability of LEID (Fig. 3g) ” (Page 14 ~ 15, Line 239 ~ 246)

Comment 3

(I.3) P.5 Fig.1 (or P.6 L 82). The authors propose an interesting solid-state electrolyte. However, the chemical nature of the reagents is not sufficiently justified. In particular the nature of the ionic liquids, why not use a more conventional electrolyte? The comparison with other salts would be an interesting way (however, I do not ask the authors to carry out the measurements, because it requires too much work). In the absence of additional measurements, it would be necessary to justify the choice of ionic liquid with State-Of-The-Art support. Concerning the epoxy precursor, it also needs more justification, why not a simple PEG? Without these elements of justification, the article is weakened, especially with regard to the proposed generic journal.

Response:

We greatly appreciate your comments. The choice of the chemical components is based on the following reasons:

(1) In order to obtain a solid electrolyte with high mechanical strength for LEID, a mechanically strong polymer matrix must be used. The conventional thermoplastic polymer, including PEO (or PEG for low molecular analog) and PVDF, are not strong enough for LEID. For example, the modulus and strength of PEO are only 3.9 MPa and 75.3 MPa , respectively [Nanotechnology, 2007, 18, 125606], which cannot meet the requirement of load-bearing applications. Therefore, we had to choose thermosetting polymers. Those polymers have high mechanical strength and are widely used in carbon-fiber-enforced polymer composite materials. The 5284 epoxy resin is an aviation-grade resin with cyanate resin as the curing agent. The bisphenol A type cyanate resin has a rigid molecular structure, thereby bringing a strong cross-linking network into the cured resin, making the resin show outstanding mechanical strength and thermal stability. [Chinese J. Aeronaut., 2015, 28(3) 903–913]. Therefore, we chose it as the matrix for our solid electrolyte.

(2) However, the thermosetting polymer has a rigid network molecular structure and small free volume after crosslinking; consequently, the transport of ions through the polymer matrix is difficult.

If LiTFSI was directly added to the epoxy resin, the conductivity would be too low to be measured. Therefore, we used a phase separation structure to solve this problem. In this structure, liquid electrolytes form an individual phase inside the porous polymer matrix, so the ion transport is not restricted by polymer chains. In the strategy, a liquid electrolyte that can phase-separate spontaneously with epoxy resin is preferred. EMIm-TFSI has been reported to create a pore in epoxy resin through in-situ phase separation during curing [J. Phys. Chem. C, 2014, 118, 49, 28377–28387], and it has good ion conductivity. Thus, we used EMIm-TFSI as the electrolyte. Besides, as an ion liquid, EMIm-TFSI is non-volatile, so our device can be stored in the air without encapsulation.

(3) Lithium salt was added into electrolyte for two purposes: firstly, lithium ion have a catalytic effect on the curing of epoxy resins and can be finished at a lower temperature [Solid State Ion, 2018, 326, 150–158; Mater. Sci. Eng. B, 2017, 219, 37–44]; secondly, lithium can improve the capacitance of the LEID by introducing lithium ion adsorption on electrodes. In fact, we have performed some experiments to help us find the optimum combination of ion liquid and the lithium salt. As shown below, EMIm-TFSI / LiTFSI has the highest conductivity. The same anion TFSI⁻ is important for the solubility of lithium salt in EMIm-TFSI. Therefore, we chose EMIm-TFSI / LiTFSI as the liquid electrolyte.

Table R3. Conductivity of different IL-based electrolyte

Electrolyte sample	Solubility	Ionic conductivity / mS cm ⁻¹ (25 °C)	Sample
EMIm-TFSI / LiCl - 1.0 mol	Insoluble	-	#1
EMIm-TFSI / LiCO ₃ - 1.0 mol	Insoluble	-	#2
EMIm-TFSI / LiTFSI - 1.0 mol	Soluble	2.9	#3
P1,3- TFSI/ LiTFSI - 1.0 mol	Soluble	2.0	#4
BMIm-TFSI / LiTFSI - 1.0 mol	Soluble	1.7	#5

Figure R1. Photos of different electrolytes.

A paragraph was added into the revised manuscript to explain the design of materials:

“High-strength solid electrolyte is essential for LEID, but the conventional solid electrolyte based on thermoplastic polymers, such as poly(ethylene oxide) (PEO) and poly(vinylidene difluoride) (PVDF), do not have sufficient high mechanical strength^{34, 35}. Therefore, solid electrolytes based on high-strength thermosetting polymers become suitable candidates. However, the thermosetting polymer has a rigid molecular chain network and small free volume after crosslinking; thus, the transport of ions through the polymer matrix is difficult. Consequently, the solid electrolyte should be designed as a bi-continuous-phase type structure, where the liquid electrolyte forms a separated phase inside the porous polymer matrix, so that the ion transport is not restricted by polymer chains. Here in this work, we developed a new bi-continuous-phase type solid electrolyte³⁶. Epoxy resin 5284 was chosen as the matrix phase, while the solution of bis(trifluoromethane)sulfonamide lithium (LiTFSI) in 1-ethyl-3-methylimidazolium bis(trifluoromethanesulfonyl)imide (EMIm-TFSI) ionic liquid was used as the conductive phase (Fig. 1a). Epoxy resin 5284 is a single-component epoxy resin with cyanate resin as the curing agent³⁷⁻³⁹. The bisphenol A type cyanate resin has a rigid molecular structure, thereby bringing a strong cross-linking network into the cured resin, making the resin show outstanding mechanical strength and thermal stability. Therefore, epoxy resin 5284 is expected to provide high mechanical properties for solid electrolytes. EMIm-TFSI was used as the electrolyte because it was

reported to undergo a spontaneous phase-separation during the curing of epoxy resin¹⁴. The solution of LiTFSI in EMIm-TFSI (LE) has high ionic conductivity (2.1 mS cm⁻¹), comparable to the commercial liquid electrolyte in the lithium-ion battery, so it can ensure a good conductivity of the solid electrolyte (Supplementary Table 2) ” (Page 6 ~ 7, Line 79 ~ 99,)

Comment 4

(I.4) P.8 L 132- 145. A more serious comparison is needed, measuring the ionic conduction of ionic liquid, LE. As well as a comparison with the State-of-The-Art. Again, a table in the Supplementary information would give credibility to the study.

Response:

As you suggested, we measured the conductivity of LE and compared it with other commercial liquid electrolytes commonly used in lithium-ion batteries. The ionic conductivity of LE is 2.1 mS cm⁻¹, comparable to those of other liquid electrolytes (**Supplementary Table 2**). Good conductivity is one of the reasons we chose it to prepare the solid electrode. The conductivity of the solid electrolyte EP₅₀ (0.67 mS cm⁻¹) is about 1/3 of that of the conductivity of LE. Such a ratio is rational because in EP₅₀ LE can only conduct ions inside the pores of the epoxy resin matrix.

The corresponding paragraph was revised:

“...EMIm-TFSI was used as the electrolyte because it was reported to undergo a spontaneous phase-separation during the curing of epoxy resin¹⁴. The solution of LiTFSI in EMIm-TFSI (LE) has high ionic conductivity (2.1 mS cm⁻¹) comparable to the commercial liquid electrolyte in the lithium-ion batteries, so it can ensure a good conductivity of the solid electrolyte (Supplementary Table 2).” (Page 7, Line 95 ~ 99)

Supplementary Table2. Conductivity comparison of ionic liquid electrolytes and organic electrolytes.

Electrolyte	Ionic conductivity / mS cm⁻¹
Ethyl methyl carbonate (EMC) / 1 M LiTFSI	3.4
Dimethyl carbonate (DMC) / 1 M LiTFSI	4.1
Diethyl carbonate (DEC) / 1 M LiTFSI	2.5
Propylene carbonate (PC) / 1 M LiTFSI	5.5
EMIm-TFSI / 1 M LiTFSI	2.9
EMIm-TFSI / 2.3 M LiTFSI	2.1
EC:DMC =1 : 1 vol% / 1 M LiPF₆	7.9

Comment 5

(II) Electrochemical measurements. Important weaknesses are identified in the present form of the manuscript. Here the list, without the metrics asked, the results are not enough strong.

(II.1) The authors should definitely perform a Ragon diagram in to position the present work with the State-Of-The-Art. This is essential.

Response:

We greatly appreciate your suggestion. Ragon diagram was added to the revised Supplementary information. However, there is little data on areal energy density and power density in the literature, so we mainly focused on comparing the gravimetric and volumetric energy and power density.

Supplementary Fig. 13 a-c Ragone plot of LEID-3 and some solid devices in the literature^{2, 11, 15, 17-23}. Based on device¹⁴: the energy and power densities were calculated based on the mass or volume of the whole device, including the CF electrodes, GF separator, and solid electrolyte. Based on GF electrodes^{11, 15, 18}: the energy and power densities were calculated based on the mass or volume of the two GF electrodes. (Supplementary Page 18, Line 153)

“The energy density and power density are calculated to be $0.13\ Wh\ m^{-2}$ ($3.4 \times 10^{-2}\ Wh\ kg^{-1}$, $144.9\ Wh\ cm^{-3}$) and $1.3\ W\ m^{-2}$ ($0.34\ W\ kg^{-1}$, $1453.3\ W\ cm^{-3}$). The Ragone plot (Supplementary Figure 13) shows that the energy densities of LEID-3 are higher than other solid devices with similar structures and electrode materials in the literature. These data are even higher than those of devices using electroactive material-modified CF as electrodes⁶. The high energy density is ascribed to the large capacitance and high working voltage of LEID-3.” (Page 14 ~ 15, Line 239 ~ 244)

Comment 6

(II.2) The authors must, without negotiation, make a self-discharge measurement, as self-discharge is a limiting parameter for capacitor technology.

Response:

Thank you for your valuable suggestion. We measured the self-discharge behavior of LEID-3. The capacitor was charged for 1800 seconds at a constant voltage of 2 V, and the self-discharge curve was recorded. As shown in **Supplementary Figure 11**, after 50,000 seconds, the voltage retention was 22.4%.

Supplementary Fig. 11 Self-discharge curve of LEID-3.

“The device was charged to 2 V at a current density of 0.16 mA cm^{-2} and then disconnected from the test circuit to rest for 20 min, before the open circuit voltage change was begun to record.” (Supplementary Page 16, Line 146 ~ 147)

“The self-discharge performance was also measured. After being charged to 2 V, the decreasing curve of the open-circuit voltage of LEID-3 was recorded. After 50000 s, the voltage dropped to 0.12 V, 22.4% of the original value. This slow self-discharge rate is similar to other supercapacitors in the literature⁵¹⁻⁵³ and demonstrates the long-term energy storage capability of LEIDs in practical applications. The self-discharge result also confirms that in LEID-3 there is no side reaction that causes obvious self-discharge.” (Page 15, Line 249 ~ 254)

Comment 7

(II.3) Authors should perform a stability measurement over more cycles (minimum 10,000 cycles) currently, they show 5,000 cycles.

Response:

According to the your suggestion, we tested the cyclic stability over 10,000 cycles. After 10000 charge-discharge cycles at a current density of 1.2 mA cm^{-2} , the capacity retention rate was as high as 92 %.

Fig. 3g was replaced by a new one:

Fig. 3 Performance of supercapacitor with three-layered structure. ... g Cyclic stability of LEID-3 in 10000 cycles at a current density of 1.2 mA m^{-2} . Inset shows the GCD curves of different cycles ...

Comment 8

(II.4) The authors make measurements within a cell-voltage limit close to 2 Volts, why? In view of the salts used, a voltage of 3-4 Volts should be accessible. A measurement of the maximum voltage would then be necessary to realize the extent of the interest of the proposed materials. Note that the voltage impacts both the energy density and the max power. Without a more rigorous measurement of this cell voltage, the article loses a major interest. Moreover, on P.14 Fig. 4, the authors show figure b, c at 2 volts and figure f and g at 4 volts. This is not well justified.

Response:

We appreciate your suggestions and fully agree with the opinion of the referee that higher voltage is desired for high energy and power density. Actually, we have investigated the voltage of the device, and the GCD curves at different voltages are shown in Figure R2. With the increase of cell voltage, the Coulomb efficiency dwindled down, especially at the small current densities. Obviously, there are side reactions at high voltage, although the electrolyte may work at 3 V. The possible reason is that considering the practical way of using LEID, we carried out all the electrochemical tests in air, where oxygen and water may be involved in the electrode process at high voltage. Therefore, currently, we chose a relatively low working voltage.

In Figure 4, the voltage scales are different because some of the devices were tested in series. The devices in Figure 4 b&c are single devices or two devices in parallel, so the working voltage was 2 V. In Figure 4 f&g, the blue curves are corresponding to two LEID-3 devices in series, so the working voltage was 4 V.

Figure R2. a-b CV (a) and GCD (b) curves at different voltages.

Comment 9

(II.5) The authors normalize their measurements per area, whereas normalization per mass and volume would be necessary.

Response:

According to your comment, we added the gravimetric and volumetric specific capacitance of the LEID-3 device in the Supplementary information (Supplementary Figure 12). It should be noted that the gravimetric performance was calculated both based on the electrode mass and total mass of the device, including the mass of the electrolyte and separator. Also, the volumetric performance was calculated based on the total volume of the device.

A figure and corresponding discussion were added to the Supplementary information and main text:

Supplementary Fig. 12 a-b Specific capacitance of LEID-3. Based on device: the specific capacitance was calculated based on the mass or volume of the whole device, including the CF electrodes, GF separator, and solid electrolyte. Based on GF electrodes: the specific capacitance was calculated based on the mass or volume of the two GF electrodes.

“According to the GCD curves, it can be calculated that at a current density of 0.2 mA cm^{-2} , the areal specific capacitance of LEID-3 is 32.4 mF cm^{-2} (Fig. 3h, or 675.9 mF g^{-1} , $1297.8 \text{ mF cm}^{-3}$, see Supplementary Figure 12), surpassing those of other solid electrochemical capacitors with CF as the electrode (Supplementary Table 5, Supplementary Figure 12). ...The energy density and power density are calculated to be 0.13 Wh m^{-2} ($3.4 \times 10^{-2} \text{ Wh kg}^{-1}$, 144.9 Wh cm^{-3}) and 1.3 W m^{-2} (0.34 W kg^{-1} , 1453.3 W cm^{-3}). The Ragone plot (Supplementary Fig. 13) shows that the energy densities of LEID-3 are higher than other solid devices with similar structures and electrode materials in the literature.

These data are even higher than those of devices using electroactive material-modified CF as electrodes⁶. The high energy density is ascribed to the large capacitance and high working voltage of LEID-3. The cycle stability of the device was tested at a current density of 1.2 mA cm⁻². The capacitance retention is 92% after 10,000 cycles, reflecting the excellent cycle stability of LEID (Fig. 3g).” (Page 14 ~ 15, Line 231 ~ 246)

Comment 10

(III) The present work, based on a LEID technology, is interesting, however, it would be appreciated if in the introduction the current or envisaged applications were a little more detailed. Nature Communication is a generalist journal with a wide readership.

Response:

Thank you for your valuable suggestion. We added a paragraph to the introduction to explain the application of LEIDs:

“...it can replace the structural component and effectively increase the total energy density of the system by reducing dead mass^{14,15}. With LEIDs as shells, satellites, electric vehicles, drones, laptops, and mobile phones can have more electric power and longer battery life. Besides, LEIDs can also serve as support structures and energy storage units for intermittent new energy sources, such as wind power and photovoltaics. Consequently, high-strength LEID significantly increases the energy density of mobile energy storage systems and simplifies the system¹⁶.” (Page 3, Line 36 ~ 40)

Comment 11 & 12 & 13

(IV) Minor remarks:

(IV.1) P.1 L.15. The description of the state of the art in the introduction is too short, a sentence on the importance of technology is needed.

(IV.2) P.1 L. 17. The authors write « high-performance », this term is too empty, it is necessary to be more precise, values should be given.

(IV.3) P.2 L 27. The authors should mention the values of energy density and maximum power obtained.

Response:

The above three comments are about the abstract (P. 1 ~ 2). According to your suggestion, we revised the abstract. The first sentence was expanded, and values of the mechanical/energy storage performance were added to the revised abstract.

“Load bearing/energy storage integrated devices (LEIDs) allow using structural parts to store energy, and thus become a promising solution to boost the overall energy density of mobile energy storage systems, such as electric cars and drones. Herein, with a new high-strength solid electrolyte, practical high-performance load-bearing/energy storage integrated electrochemical capacitors with excellent mechanical strength (flexural modulus: 18.1 GPa, flexural strength: 160.0 MPa) and high energy storage ability (specific capacitance: 32.4 mF cm⁻², energy density: 0.13 Wh m⁻², maximum power density: 1.3 W m⁻²) are reported. For the first time, two basic types of multilayered structures are introduced into LEID, and they are proven to significantly enhance the practical bearing ability and working flexibility of the device. Besides, the device can be made into complex structures composed of curved and planar surfaces, and secondarily machined and assembled without affecting their energy storage ability, demonstrating its excellent processability.” (Page 1 ~ 2, Line 12 ~ 22)

Comment 14

(IV.4) P.4 L 61. A comparison with the state-of-the-art on solid-state-electrolyte would be appreciated.

Response:

The state-of-the-art on solid-state-electrolyte was shown in Fig.1h in the original manuscript. In order to show the comparison more clearly, we added a table in Supplementary information:

Supplementary Table 3. The Ionic conductivity and mechanical strength comparison of the solid electrolytes in literature with EP₅₀ in this work.

Electrolyte	Ionic conductivity / mS cm ⁻¹	Young's modulus / GPa	Reference
MVR444 / EMIm-TFSI / LiTFSI	0.007	0.81	14
Epoxy / SN / LiTFSI	0.35	0.3	29
CCR Epoxy / BMIBF ₄ / LiBF ₄	0.00135	0.045	30
Epoxy / PEG / LiTF	0.0086	0.135	31
DGEBA Epoxy / SN / LiTFSI	0.2	0.001	32
DGEBA Epoxy / BMIm-TFSI/ LiTFSI	0.26	0.5	33
EP ₅₀ in this work	0.67	1	-

(Supplementary Page 33, Line 204 ~ 205)

Comment 15

(IV.5) P.9 L 153. In the title the authors note LEID-3. This should be avoided, especially the 3, which refers to an internal notation and not an internationally accepted nomenclature.

Response:

Thank you for your suggestion. The title was revised, and other titles with the same problem were also revised:

“Fabrication and characterization of the three-layer load-bearing/energy-storage integrated device.”

(Page 11, Line 167 ~ 168)

“Design, fabrication, and performance of multilayer load-bearing/energy-storage integrated device.”

(Page 17, Line 272~ 273)

Comment 16

(IV.6) SI P.2 L18. The composition in the table is not sufficiently explained.

Response:

Following the suggestion, we revised the table to make it easy to read. Now the table shows the formula of the different solid electrolytes.

Supplementary Table 1. The composition of the precursor mixture for different electrolytes.

sample	EP5284 : EMIm-TFSI : LiTFSI : PC (by weight)
EP₁₀₀	100 : 0 : 0 : 0.
EP₇₀	70 : 30 : 13 : 0.3
EP₆₀	60 : 40 : 17.4 : 0.4
EP₅₅	55 : 45 : 19.5 : 0.45
EP₅₀	50 : 50 : 21.8 : 0.5
EP₄₅	45 : 55 : 23.9 : 0.55
EP₄₀	40 : 60 : 26.1 : 0.6

Referee #2

Zhang, et al., “Machinable multilayered load-bearing/energy storage integrated electrochemical capacitor based on high-performance polymeric solid electrolyte”. This paper presents the polymeric solid electrolytes with both mechanical strength and ionic conductivity for the development of load bearing/energy storage integrated devices (LEIDs). The authors prepared a series of epoxy resin-based solid electrolytes containing LiTFSI dissolved in EMIm-TFSI ionic liquid via the thermal curing process, and investigated their ionic conductivity and mechanical properties, combined with the morphological study. Then, the authors designed the multilayered LEIDs based on the electrolyte, CF, and GF, and explored their electrochemical performance under mechanical and machinable deformation. These results could be useful to the readers who are interested in the development of multifunctional structural devices with both mechanical bearing capacity and electrochemical storage capacity, but there still exist questionable points that need to be addressed before this paper can be published, detailed below.

Response:

Thank you very much for your constructive comments on our manuscript. We had revised the manuscript following your suggestions, and the revisions we made are described below.

Comment 1

1. Page 5, Fig. 1a: In Fig. 1a, “LiTSFI” should be replaced with “LiTFSI”.

Response:

The error in Figure 1a has been corrected. We also carefully checked the manuscript and corrected all the typos we found.

Fig. 1 Structure and properties of the epoxy resin-based solid electrolyte. **a** Schematic diagram of the preparation of solid electrolyte with epoxy resin and ionic liquid, and the chemical structures of the epoxy resin and the ionic liquid. **b** Photos of epoxy resin with different ionic liquid content. **c**, **d** SEM images of EP₇₀ (**c**) and EP₅₀ (**d**). **e** Tensile stress-strain curves of epoxy resin and different solid electrolytes. **f** Impedance spectra of different solid electrolytes. **g** The Flexural modulus (left axis) and ionic conductivity (right axis) of different samples at 25 °C. **h** Comparison of the solid electrolytes in literature with EP₅₀ in this work^{14, 29-33}. (Page 6, Line 72 ~ 78)

Comment 2

Page 5, Lines 95-96 & Page 21, Lines 371-375: To determine the curing temperature, the authors conducted the DSC measurement (SI, Fig. 1). The authors also mentioned that “As the content of ionic liquid (IL) increases, the exothermic peak shifts to lower temperature.”. However, the peak temperature seems to have a non-monotonic IL concentration dependence; i.e., the EP50 with the lower IL content has a lower peak temperature (117 °C) than the EP40 with the higher IL content (123 °C). Can the authors explain such a non-monotonic dependence? in addition, how about the correlation between the IL content and the enthalpy of the exotherms (ΔH)? In the section of “Preparation of solid electrolyte”, the authors mentioned that the solid electrolyte was pre-cured at 100 °C for 8 min. Is there any specific reason for performing the pre-curing process at 100 °C? Although the exothermic peak of the EP prepolymer was observed at 275 °C from the DSC curve (SI, Fig 1), it was cured at 160, 170, and 180 °C, not at ~275 °C. Can the authors also explain this point ?

Response:

(1) Thank you very much for your valuable comments. Our description in the original manuscript was not accurate. We carefully checked the original data and repeated the DSC experiments. It was found that the peaks shifted to ~133 °C when LiTFSI was added, but there were minor deviations between each measurement result, which caused the non-monotonic change of the peak temperature. These deviations originate from the random fluctuation of the equipment and the preparation of the samples. Therefore, the conclusion should be that LiTFSI can significantly reduce the maximum exothermic peak temperature to around 133 °C, but it is difficult to confirm the dependence of the peak temperature on IL concentration. Therefore, we revised the corresponding paragraph:

“Supplementary Fig. 1 shows that the maximum exothermic peak temperature of various samples containing lithium salt is around 133 °C, but the maximum exothermic peak temperature of pure epoxy resin is ~ 275 °C. Therefore, lithium salt has a catalytic

effect on the solidification of the polymer electrolytes¹⁻⁵.” (Supplementary Page 4, Line 56 ~ 59)

(2) As you suggested, we calculated the enthalpy of the curing reaction according to the DSC curves. The results are shown in **Supplementary Fig. 1**. The enthalpy decreased significantly as LE (liquid electrolyte) was added, indicating that the crosslinking density was reduced. This change can be explained by the fact that with the addition of LE, the concentration of epoxy resin prepolymer decrease, and consequently, the crosslinking probability decrease. The results and corresponding discussion were added to the revised main text and the Supplementary.

Supplementary Fig. 1 DSC measurement of precursors of EP₁₀₀, EP₇₀, EP₆₀, EP₅₀, and EP₄₀. **a** DSC curves. **b** Low-temperature part of the curves in (a). **c** *Mathematical integration plot of the heat flux for the calculation of enthalpy.* **d** *Enthalpy change (ΔH) of the curing reaction. Calculated values were obtained from the ΔH of pure epoxy resin and the content of epoxy resin in the solid electrolytes (neglecting the change of heat capacity).*

“The enthalpy of the curing reaction was calculated according to the DSC curves, as shown in Supplementary Fig. 1. The curing enthalpy of pure epoxy resin was 658.7 kJ g⁻¹, and when LE was added, the curing enthalpy of the mixture was expected to decrease. The measured enthalpy was smaller than the calculated value, indicating that the crosslinking density was reduced. With the addition of LE, the concentration of epoxy resin prepolymer decreased, and consequently, the crosslinking probability decreased.” (Supplementary Page 4 ~ 5, Line 60 ~ 65)

“The curing process was first investigated by differential scanning calorimetry (DSC, Supplementary Fig. 1) and gel point tests (Supplementary Fig. 3) to determine the curing temperature, and the successful curing was finally verified by Fourier transform infrared (FT-IR) spectra (Supplementary Fig.4). However, the measurement of the curing enthalpy suggests that the crosslinking density of epoxy resin decreased after LE was added (Supplementary Fig.1), because LE reduced the concentration of epoxy groups.” (Page 7, Line 106 ~ 111)

(3) The purpose of pre-curing at 100 °C is to pre-gelate the prepolymer on the surface of the electrode. Pre-curing can increase the viscosity of the prepolymer to prevent leakage during the following hot-pressing and was also used in the literature [Compos Sci Technol, 2011, 71, 765–771; Compos. Commun., 2020, , 21, 100401]. The temperature of 100 °C was chosen because, at this temperature, the viscosity of our precursor can reach the desired level within a suitable time (8 min). The reason for pre-curing was added in the revised manuscript:

“The solid electrolyte precursor mixture was pre-cured at 100 °C for 8 min to increase the viscosity and avoid leakage during in following procedures, and then transferred into the mold and cured using temperature programming” (Page 24, Line 407 ~ 408)

(4) The curing of epoxy resin is a spontaneous reaction that occurs even at room temperature, although the reaction is very slow near room temperature. Practically, the

resin curing is usually carried out at a higher temperature to increase the reaction rate. However, we usually do not cure the resin at the measured exothermic peak temperature (T_p), because the exothermic curing reaction will be too fast, and the reaction heat may damage the resin.

Here we also measured the intrinsic exothermic peak temperature (T_i) using the extrapolation method [Materials, 2021, 14, 4673]. The peak temperature increases with the heating rate, so the T_i can be obtained when the heating rate is extrapolated to zero. The DSC curves of 5284 epoxy resin at heating rates of 5, 10, 15, and 20 °C min⁻¹, and the extrapolation result are shown in Figure R3. From the curve shown, it can be found that the T_i is 240.7 °C. In our experiments, the curing temperature was set at 180 °C, about 60 °C lower than the T_i . This temperature ensured a high curing rate and crosslinking density and avoided overheating.

Figure R3. **a** DSC curves of epoxy precursors with different heating rates. **b** Extrapolation of exothermic peak temperature to obtain the intrinsic maximum exothermic peak temperature.

Comment 3

Page 7, Lines 104-107 & Lines 116-117: The authors mentioned that “the solid electrolytes have a bi-continuous structure composed of a LE phase and a LE-plasticized epoxy resin phase.”. If the solid electrolytes have the bi-continuous structure, then the bi-continuous electrolytes would be expected to have two glass transition

temperatures (T_g s). However, from the DMA measurement (SI, Fig 5), it looks like the electrolytes have a single T_g , which is presumably attributed to the LE-plasticized epoxy phase. What about the T_g from the LE phase of the solid electrolyte? DSC measurements may be useful to observe the LE-phase T_g at lower temperatures.

Response:

Thank you for your valuable suggestion. Following your suggestion, we supplemented the low-temperature DSC experiment from -175 °C to 200 °C. The result shows that there is indeed another T_g in the low-temperature region at -73 °C, while the other T_g is at 85 °C. The first one should be related to the LE phase, and the second one can be ascribed to the epoxy resin phase plasticized by the LE. The T_g measured by DSC is lower than that by DMA because of the differences in testing methods [Polym. Chem., 2016, 7, 3071; J. Mater. Sci., 2021, 56, 936–956].

A new figure was added to the revised Supplementary Information, and a sentence was added to the main text:

“.....(Supplementary Fig.5), showing that phase separation between LE and epoxy occurred. This phase separation is further confirmed by the DSC measurement (Supplementary Fig. 2), in which two glass transition temperatures were observed.”

(Page 8, line 122 ~ 123)

Supplementary Fig. 2 DSC test of cured EP₅₀.

“Two glass transition temperatures at -73 °C and 85 °C were observed on the DSC curve. The first one should be related to the LE phase, and the second one can be

ascribed to the epoxy resin phase plasticized by the LE.” (Supplementary Page 6, line 68 ~ 70)

Comment 4

Page 8, Lines 132-133: The authors mentioned that “the ionic conductivity of the solid electrolyte increased with the LE” (Fig. 1f, g and SI, Fig. 7). However, in SI Fig. 7 (Nyquist plot), the EP45 seems to have lower resistance than the EP40, which is not consistent with the result shown in Fig. 1g. The authors should re-check this conductivity result.

Response:

Thank you for your careful review. In Fig. S7 (S8 in the revised Supplementary information), EP₄₅ indeed has lower resistance than the EP₄₀, because the sizes of the EP₄₅ and EP₄₀ are different. The resistance of EP₄₀ sample is 9.9 Ω , and the area and the thickness are 15.2 cm² and 0.142 cm, respectively. Therefore, the conductivity is 0.95 mS cm⁻¹. For EP₄₅, the resistance, area, and thickness of EP₄₅ are 7.9 Ω , 20.25 cm², and 0.136 cm, respectively. These data lead to a lower conductivity of 0.7 mS cm⁻¹. Here we used the measured impedance values and the sizes of the samples to calculate a formal resistivity. In the Nyquist plot of formal resistivity, we are easy to find out that EP₄₀ has lower resistivity than EP₄₅.

Supplementary Fig 8 was revised to provide a better view of the Nyquist plot:

Supplementary Fig. 8 a-b Enlarged view of impedance spectra of different solid electrolytes between block electrodes. **c-d** The formal resistivity (ρ) calculated from data in (a) and (b). ($\rho=ZA/d$, where A and d are the area and thickness of the electrolyte sample.) (Supplementary Page 13, line 126 ~ 128)

Comment 5

Page 8, Lines 143-145: The references cited in the sentence seem to be different from those in Fig. 1h. If so, the authors should ensure that all the references are correctly cited in this manuscript.

Response:

Thank you for your careful review. We carefully checked the reference and corrected all the errors.

“Figure 1h compares the performance of EP₅₀ in this work and the literature data, and the data point of EP₅₀ is located in the upper right corner of the figure, indicating that its comprehensive performance is better than the literature value^{14,29-33}.” (Page 10, Line 158 ~ 160)

Comment 6

Page 10, Line 157: What is GF? Is it glass fiber? There is no information about GF, such as thickness, pore size, etc., in this manuscript. For the LEID fabrication, the authors seemed to use GF as the separators. Even though the solid electrolyte itself can act as a separator, why did the authors use GF?

Response:

Thank you for your suggestion. GF here is glass fiber. The GF with model SW280F-90a in this work is the product of Nanjing Fiberglass Research and Design Institute Co., Ltd. The areal density of the GF fabric is $280 \pm 20 \text{ g cm}^{-2}$, and the thickness is $0.250 \pm 0.025 \text{ mm}$. The diameter of single glass fiber is $8 \pm 0.5 \text{ }\mu\text{m}$. This information was added in the experimental part.

While the solid electrolyte layer can be used as a separator, in our experiment, the electrolyte was prepared by in-situ curing, which means that it is fluid when the LEID is assembled. Therefore, using only solid electrolytes as the separator may lead to short circuits between the adjacent two CF electrodes. GF is insulating and commonly used as the reinforcement in composite materials, so we used it as the separator to avoid short circuits.

Following sentences were added to the revised manuscript

“As shown in Fig. 2a, CF fabrics were used as the electrodes, and glass fiber (GF) fabrics as the separators to avoid short circuits, and both CF and GF fabrics also served as the mechanical reinforcements.” (Page 11, Line 170 ~ 172)

“The GF (SW280F-90a) was bought from Nanjing Fiberglass Research and Design Institute Co., Ltd. The areal density of the GF fabric is $280 \pm 20 \text{ g cm}^{-2}$, and the thickness is $0.250 \pm 0.025 \text{ mm}$. The diameter of single glass fiber is $8 \pm 0.5 \text{ }\mu\text{m}$.” (Page 23, Line 396 ~ 398)

Comment 7

Page 10, Lines 163-164, SI Fig 9b: For the electrochemical performance of the LEID-3 car shell in SI Fig. 9b, the GCD curve at the current density of 0.04 mA cm^{-2} is missing. Please revise the figure.

Response:

Thank you for your careful review. The legend of Supplementary Fig. 10 was revised:

Supplementary Fig. 10 a the CV curves of the automobile model shell from 10 mV s^{-1} to 100 mV s^{-1} . **b** GCD curves of the automobile model shell. **c** Impedance spectra of the automobile model shell. **d - h** The photos of the automobile model shell powering LED lights under different loads.

(Supplementary Page 15, line 137 ~ 140)

Comment 8

Page 12, Line 202 & Page 17, Line 303: There are several typos in the numbers of the main and SI figures; for example, “Fig. 3d” should be replaced with “Fig. 3c” and “Supplementary Fig. 16” should be replaced with “Supplementary Fig. 17”. The authors should re-check all the figure numbers.

Response:

Thank you for your valuable suggestion. We have carefully checked the figure numbers and corrected all the errors.

Comment 9

Page 13, Lines 213-214 & Fig 3e: The authors mentioned that “the current densities of the CV curves are proportional to the scanning rate showing a typical capacitive behavior”. The authors can directly prove it by plotting current vs. scan rate and by analyzing the slope of the linear dependence.

Response:

Thank you for your valuable suggestion. We tried to linearly fit the current with the scan rate but found that although the current increased with the scan rate, the linearity was poor. One important reason is that the current did not reach a steady state because of the large series resistance. Therefore, we changed the description in the revised manuscript:

“The cyclic voltammetry (CV) curves at different scanning rates in the voltage range of 0 ~ 2 V all have a deformed rectangular shape without pronounced redox peaks, showing a typical capacitive behavior (Fig. 3e).” (Page 14, Line 227 ~ 229)

Comment 10

Page 13, Lines 216-218: The authors indicated the calculated areal specific capacitance, energy density, and power density of LEID. The comparison of this device with the state of the art should also be presented in Ragone plot and discussed in the text.

Response:

Thank you for your valuable suggestion. Following your suggestion, we compare the specific capacitance (by mass, area, and volume) (**Supplementary Table 5**, **Supplementary Fig. 12**) of LEID-3 with other solid devices based on similar electrode materials. The specific capacitance of LEID-3 is much higher than that of devices with CF as the electrode in the literature and comparable to some devices with electroactive electrode materials.

We also compared the energy density and power density (**Supplementary Fig. 13**) of LEID-3 with the literature data. However, there is little data on areal energy density and power density in the literature, so we mainly focused on comparing the gravimetric and volumetric energy and power density. **Supplementary Fig. 13** shows that our energy densities are higher than the literature data and are even higher than those of devices using electroactive material-modified CF as electrodes. The high energy density is ascribed to the large capacitance and high working voltage. The above data demonstrate that our LEIDs have good energy storage performance comparable with other solid devices in the literature.

A table and two figures were added to the Supplementary information, and a paragraph was added to the main text:

Supplementary Table 5. Comparison of the specific capacitance of LEID in this work with other solid-state supercapacitors in the literature.

Electrode	Separator	Electrolyte	Specific capacitance	Reference
CF	GF	EP-IL	$C_a=32.4 \text{ mF cm}^{-2b}$ $C_v=1297.8 \text{ mF cm}^{-3b}$ $C_g=675.9 \text{ mF g}^{-1b}$ $C_a=5.68 \text{ mF cm}^{-2b}$	This work
MnO ₂ -CF	GF	EP-IL	$C_v=82 \text{ mF cm}^{-3b}$ $C_g=49 \text{ mF g}^{-1b}$	24
Vertical Graphene/Mn O ₂ - CF	GF	EP-IL	$C_v=30.7 \text{ mF cm}^{-2b}$	19
CuO-CF	GF	Polyester - LiTf- EMIMBF ₄	$C_g=6.75 \text{ F g}^{-1a}$	15
Activated CF	FP	EP-TEABF ₄	$C_g=25.4 \text{ mF g}^{-1b}$	25
Carbon aerogel -CF	GF	PEGDGE-IL	$C_a=3.15 \text{ mF cm}^{-2b}$ $C_v=34.6 \text{ mF cm}^{-3b}$ $C_g=71.2 \text{ mF g}^{-1b}$	2
Graphene nanoplatelet CF	- FP	DGEBA- LiClO ₄	$C_v=118.7 \text{ mF cm}^{-3b}$	23
Urea- Activated Grap hene-CF	GF	PEGDGE-IL	$C_v=82.3 \text{ mF cm}^{-3b}$	17
ZnO-CF	GF	Polyester - LiTf- EMIMBF ₄	$C_g=10.6 \text{ F g}^{-1a}$	11
Cu-Co-Se-CF	KF	Polyester - LiTf- EMIMBF ₄	$C_g=28.6 \text{ F g}^{-1a}$	18
MWCNTs-CF	GF	PEG-LiTf	$C_g=125 \text{ mF g}^{-1c}$	26

^a Specific capacitance was calculated based on the mass of active materials. (without the mass of CF and electrolyte)

^b Specific capacitance was calculated based on the mass of electrodes. (The mass of CF or total mass of active materials and CF)

^c Specific capacitance was calculated based on the mass of device. (Supplementary Page 36, Line 209 ~ 213)

Supplementary Fig. 12 a-b Specific capacitance of LEID-3. Based on device: the specific capacitance was calculated based on the mass or volume of the whole device, including the CF electrodes, GF separator, and solid electrolyte. Based on GF electrodes: the specific capacitance was calculated based on the mass or volume of the two GF electrodes. (Supplementary Page 17, Line 149 ~ 152)

Supplementary Fig. 13 a-c Ragone plot of LEID-3 and some solid devices in the literature^{2, 11, 15, 17-23}. Based on device¹⁴: the energy and power densities were calculated based on the mass or volume of the whole device, including the CF electrodes, GF separator, and solid electrolyte. Based on GF electrodes^{11, 15, 18}: the energy and power densities were calculated based on the mass or volume of the two GF electrodes. (Supplementary Page 18, Line 154 ~ 158)

“According to the GCD curves, it can be calculated that at a current density of 0.2 mA cm^{-2} , the areal specific capacitance of LEID-3 is 32.4 mF cm^{-2} (Fig. 3h, or 675.9 mF g^{-1} , $1297.8 \text{ mF cm}^{-3}$, see Supplementary Figure 12), surpassing those of other solid electrochemical capacitors with CF as the electrode (Supplementary Table 5, Supplementary Figure 12). ...The energy density and power density are calculated to be 0.13 Wh m^{-2} ($3.4 \times 10^{-2} \text{ Wh kg}^{-1}$, 144.9 Wh cm^{-3}) and 1.3 W m^{-2} (0.34 W kg^{-1} , 1453.3 W cm^{-3}). The Ragone plot (Supplementary Fig. 13) shows that the energy densities of LEID-3 are higher than other solid devices with similar structures and electrode materials in the literature. These data are even higher than those of devices using electroactive material-modified CF as electrodes⁶. The high energy density is

ascribed to the large capacitance and high working voltage of LEID-3. The cycle stability of the device was tested at a current density of 1.2 mA cm^{-2} . The capacitance retention is 92% after 10,000 cycles, reflecting the excellent cycle stability of LEID (Fig. 3g) ” (Page 14 ~ 15, Line 231 ~ 246)

Comment 11

Page 18, Lines 324-326 & Fig. 5i: The authors mentioned that after the secondary processing, the new assembled sheet works well as an electrochemical capacitor, as indicated by the GCD curves in Fig. 5i. However, the GCD curves in Fig. 5i exhibit considerable IR drop, unlike the GCD curves shown in Fig. 4c. The authors should directly compare and explain the GCD curves before and after the secondary processing.

Response:

The device in Fig. 5i comprises two LEID-3 connected in series, so the voltage drop is expected to be twice that of a single device. However, we found that the voltage drop was still over-large, which was possibly due to the poor contact between the two devices in the previous measurement. Therefore, we repeated the experiment and improved the electric contact. In the new data, the voltage drop was 0.12 V, similar to the voltage drop in Fig. 4.

Fig. 5 in the manuscript was revised:

Fig. 5 Failure assessment and secondary processing of supercapacitors. *a* The photo of a drilled LEID-3 as a power source to drive a fan. *b, c* CV curves (*b*) and GCD curves (*c*) at current density of 0.5 mA cm^{-2} of LEID-3 before and after drilling. *d* The photo of a LEID-3 lighting up the LEDs after being cut into thr pieces. *e, f* CV curves (50 mV s^{-1}) (*e*) and GCD (*f*) curves at current density of $1.8 \times 10^{-2} \text{ mA cm}^{-2}$ of LEID-3 before and after being cut. *g, h* The photos of a car model with two LEID-3 plates assembled by plastic fasteners as both the chassis and the power supply. *i* GCD test of assembled automobile based on LEID-3 chassis (0.2 mA cm^{-2}).

Comment 12

Page 18-19 & Fig. 5: For the stable and rigid external impact resistive device, the higher mechanical strength and flexibility should be given. From the images of a car model shell made of LEID-3 (Fig. 2c) and of different curve shapes (SI, Fig. 8), the device seems to have flexibility. However, there is no investigation for electrochemical

performance under mechanical deformation such as capacitance retention as a function of bending radius or bending cycle numbers. Furthermore, bending can cause the local expansion/contraction of the electrodes and electrolyte. This will induce interface contact failure, especially with solid-state devices. As a result, poor contact between electrodes and electrolyte is a critical issue for solid-state electrochemical devices. Therefore, the authors should provide the cross-section image of electrode-electrolyte integration before and after the mechanical deformation. This will further demonstrate that the LEID has high electrochemical stability.

Response:

Thank you for your valuable suggestion. The LEID-3 is rigid and not a typical flexible device. The car shell showed reversible deformation under the compact because its span is large. However, according to your suggestion, we tried to measure the electrochemical properties of LEID-3 during bending. Figure 3i shows that the specific capacitance of the device did not change with the deflection in a single measurement. The above results indicate that the interface between solid electrolyte and electrode was relatively stable. We also measured the capacitance of LEID-3 during repeating bending/releasing cycles. The capacitance was measured under the bent state during the repeating bending and under the original shape after 100 bending/releasing cycles (Supplementary Figure 16). The specific capacitance under the bent state decreased gradually during continuous bending/releasing cycles but recovered to the original value immediately when the device shape was recovered. The decrease in specific capacitance is due to the volume expansion of the device during bending, which may cause poor contact between the LE phase and the CF electrode. Nevertheless, the repeating bending also did not cause irreversible damage to the device. Therefore, in practical applications, the device can withstand small elastic deformation without irreversible degradation of electrochemical performance.

According to your suggestions, we compared the cross-section of the LEID-3 before and after 100 bending tests. As shown in Supplementary Figure 16, the microstructure of the LEID-3 did not change after being bent. Firstly, the epoxy resin was plasticized

by LE, so it has improved toughness. Secondly, the pores in the solid electrolyte can deform when squeezed, thus avoiding the local crack.

A paragraph and several figures were added to the revised main text :

“We further investigated the influence of deformation on the capacitance of LEID-3. The specific capacitance of a LEID-3 device was measured when the device was under a three-point bending test. Figure 3i shows that the specific capacitance of the device did not change with the deflection. SEM images (Supplementary Fig. 15) demonstrate that the structure of the device was intact after bending. The specific capacitance of the device also remained unchanged after 100 bending/releasing cycles (Supplementary Fig. 14, 16). The above results indicate that the interface between solid electrolyte and electrode was stable under small deformation. Therefore, in practical applications, the device can withstand small elastic deformation without irreversible degradation of electrochemical performance.” (Page 15 ~ 16, Line 254 ~ 262)

Fig. 3 Performance of supercapacitor with three-layered structure. i Areal specific capacitance of LEID-3 at different bending deflections. Inset shows the photo of three-point bending test.

Supplementary Fig. 14 a-h Photos of LEID-3 under different bending deflections. i-j CV (i) and GCD (j) curves of LEID-3 under different bending deflections.

(Supplementary Page 19, Line 160 ~ 161)

Supplementary Fig. 15 a-b SEM s before bending. c-d SEM pictures after bending.

Supplementary Fig. 16 Electrochemical performance of a LEID-3 device during 100 bending/releasing cycles. a The CV curves. b The GCD curves. c Areal specific capacitance at different cycles.

“The capacitance was measured under the bent state during the repeating bending and then under the original shape after 100 bending/releasing cycles. The specific capacitance under the bent state decreased gradually during continuous bending/releasing cycles but recovered to the original value immediately when the device shape was recovered.” (Supplementary Page 21, Line 167 ~ 170)

Comment 13

Page 21: For the CF electrode used in the device, its specific surface area, mass loading, or size is not given in this manuscript. The authors should provide such information about the electrode used for capacitor evaluation.

Response:

Thank you for your valuable suggestion and careful review. According to your suggestion, we added new experimental results in the Supplementary Information (Supplementary Fig. 17). The CF in this study is model GW3031 polyacrylonitrile-based carbon fiber produced by Weihai Development Co., Ltd, China, with a diameter of 10 μm . The CF fabric is a satin fabric with an areal density of $280 \pm 20 \text{ g cm}^{-2}$ and a thickness of $0.250 \pm 0.025 \text{ mm}$. The specific surface area was measured to be $0.44 \text{ m}^2 \text{ g}^{-1}$. It is worth mentioning that CF was not modified with any active material. This information was added to the revised manuscript:

Supplementary Fig. 17 BET test of the CF. **a** N_2 adsorption/desorption isotherms. **b** corresponding pore size distributions.

“The CF fabric (model GW3031) is a satin fabric with an areal density of $280 \pm 20 \text{ g cm}^{-2}$ and the thickness of $0.250 \pm 0.025 \text{ mm}$. The specific surface area was measured to be $0.44 \text{ m}^2 \text{ g}^{-1}$, and the average pore size was 5.3 nm (Supplementary Fig. 17).”

(Page 23, Line 393 ~ 395)

REVIEWERS' COMMENTS

Reviewer #1 (Remarks to the Author):

This scientific paper is a revision of the first submission. In the first submission, the authors omitted a set of metrics that made it easier to compare the present article with state-of-the-art. This revised version offers a clearer view of the work and shows the originality and contribution of the authors. Even if the scientific aspects are partly solved, it is a pity that the critical new data (the Ragon and the self-discharge) are not included in the main manuscript but only in the supporting information. I advise the authors to revise their figures to include these changes in the main manuscript (add the Ragon and the self-discharge). With this new addition, the paper is, I believe, of high quality for publication in the prestigious journal Nature Communication.

Reviewer #2 (Remarks to the Author):

The revised manuscript well addressed the concern of the reviewer. The revised version is of high scientific quality for the publication on Nature Communications. I recommended the acceptance of this manuscript as the current form.

Point-by-Point Response to Reviewers' Comments

Review #1

Comment 1

This scientific paper is a revision of the first submission. In the first submission, the authors omitted a set of metrics that made it easier to compare the present article with state-of-the-art. This revised version offers a clearer view of the work and shows the originality and contribution of the authors. Even if the scientific aspects are partly solved, it is a pity that the critical new data (the Ragon and the self-discharge) are not included in the main manuscript but only in the supporting information. I advise the authors to revise their figures to include these changes in the main manuscript (add the Ragon and the self-discharge). With this new addition, the paper is, I believe, of high quality for publication in the prestigious journal Nature Communication.

Response:

Thank you very much for your valuable suggestions on our manuscript. Following your suggestion, we revised the manuscript and moved the Ragon plots and self-discharge curve into the main manuscript from the supporting information. In addition, because there are too many panels in the current **Fig. 3**, this figure has been split into two new figures.

Fig. 3 Mechanical properties of LEID-3. **a** Photo of a LEID-3 supporting a 10 kg weight. **b** Bending stress-strain curves of CM-3 and LEID-3. **c** Load-distance curves in interlaminar sheet test of CM-3 and LEID-3. **d** Comparison of the mechanical properties of LEID-3 and some engineering plastics. (Page 12, Line 191 ~ 196)

Fig. 4 Electrochemical properties of LEID-3. **a** CV curves of LEID-3 at different scanning rates. **b** GCD curves of LEID-3 under different current densities. Inset shows the GCD curves of different cycles. **c** Areal specific capacitance of LEID-3 at different current densities. **d, e** Ragone plots of the LEID-3 and literature data. (Based on device: the energy and power densities were calculated based on the mass or volume of the whole device, including the CF electrodes, GF separator, and solid

electrolyte. Based on CF electrodes: the energy and power densities were calculated based on the mass or volume of the two CF electrodes or total mass of active materials and CF.) **f** The photo of a LEID-3 lighting up these LED lights. **g** Cyclic stability of LEID-3 in 10,000 cycles at a current density of 1.2 mA m^{-2} . **h** Self-discharge curve of LEID-3. **i** Areal specific capacitance of LEID-3 at different bending deflections. Inset shows the Photo of three-point bending test.(Page 14, Line 223 ~ 233)

Review #2

The revised manuscript well addressed the concern of the reviewer. The revised version is of high scientific quality for the publication on Nature Communications. I recommended the acceptance of this manuscript as the current form.

Response:

Thank you for your recommendation.